# Spin-polarized edge modes between different magnet-superconductor-hybrids

Felix Zahner [1,5] ✉, Felix Nickel [2,5] ✉, Roberto Lo Conte [3], Tim Drevelow [2], Roland Wiesendanger [1], Stefan Heinze [2,4] & Kirsten von Bergmann [1] ✉

The interplay of magnetism and superconductivity can lead to intriguing emergent phenomena. Here we combine two different two-dimensional anti-ferromagnetic magnet-superconductor hybrids (MSH) and study their properties using spin-polarized scanning tunneling microscopy. Both MSHs show the characteristics of a topological nodal point superconducting phase with edge modes to the trivial substrate superconductor. At the boundary between the two MSHs we find low-energy modes which are spin-polarized. Based on a tight-binding model we can explain the experimental observations by considering two different topological nodal point superconductors. At their boundary spin-polarized chiral edge modes emerge that connect topological nodal points of the two different MSH. We demonstrate via the complex band structure that due to an asymmetric lateral decay these edge modes are spin-polarized, regardless of the details of the spin structure at the boundary. This work shows how interfaces between two distinct topological nodal point superconductors can serve as a platform to engineer spin-polarized edge modes.

Topological superconductivity has become of great interest, as it allows the formation of topologically protected states, such as Majorana bound states (MBS) and edge modes[1]. These modes can occur at the boundaries between trivial and topological superconductors[2,3] or between two different topological superconducting phases. A promising platform to establish topological superconductivity are magnet-superconductor hybrid (MSH) systems, where magnetic chains or thin layers are grown on a conventional $s$-wave superconductor[4]. In particular, two-dimensional (2D) magnets can host a plethora of different spin textures[5,6] which can be placed in direct proximity to a superconductor, emerging as an ideal material class for the search for topological superconductivity. This is confirmed by several theoretical predictions of the emergence of intriguing phenomena in such hybrid systems[7–13].

In the last decade, a variety of 2D magnet-superconductor hybrids have been investigated both experimentally and theoretically. In those previous studies the boundary between a topological and a trivial superconducting phase was investigated, as it is the case for magnetic islands on the surface of a conventional superconductor. Here, the topological phase arising due to the magnetic island is surrounded by the trivial phase of the superconducting substrate. For a ferromagnetic state[3,8,14,15] or non-coplanar spin textures[9,10,13,16,17] gapped topological superconductivity has been found or predicted. Antiferromagnetic (AFM)[12,18–20] or spin spiral[16,21] states can induce topological nodal point superconductivity (TNPSC). This TNPSC can generally be expected for AFM textures due to the combined symmetry of time-reversal and lattice translation. Antiferromagnetism-based MSHs are of particular interest for applications, because in addition to their intriguing topological superconductivity they exhibit a zero net moment and ultrafast dynamics. In contrast to gapped superconductors, such TNPSCs have at least one pair of nodal points (NP), where the band gap is closed[22–24]. The two NPs are characterized by opposite topological charges. Such

[1]Department of Physics, University of Hamburg, Jungiusstraße 11, Hamburg, Germany. [2]Institut für Theoretische Physik und Astrophysik, Christian-Albrechts-Universität zu Kiel, Kiel, Germany. [3]Zernike Institute for Advanced Materials, University of Groningen, Groningen, The Netherlands. [4]Kiel Nano, Surface, and Interface Science (KiNSIS), University of Kiel, Kiel, Germany. [5]These authors contributed equally: Felix Zahner, Felix Nickel. ✉e-mail: felix.zahner@uni-hamburg.de; nickel@physik.uni-kiel.de; kirsten.von.bergmann@uni-hamburg.de

topological NPs are protected against spontaneous annihilation and can only be detached from the Fermi surface when a pair of NPs is merged. At boundaries with a trivial superconducting phase, NPs of opposite topological charge connect via low-energy modes. The occurrence of the edge mode depends on the projection of the NPs on the crystallographic direction of the boundary[12,18,20,21], resulting in the emergence of different edge modes for non-equivalent edges of the MSH system.

Edge modes, appearing as an increased density of states at boundaries between different regions, can in-principle have a topological or non-topological origin. Recent studies have found the emergence of Yu-Shiba-Rusinov (YSR) states at edges of 2D van-der-Waals materials[25,26], originating from a different coupling to the superconducting layer at the edges compared to the interior of islands. Furthermore, Andreev bound states may form in superconductors (SCs), also giving rise to a local modification of the in-gap states at boundaries[27,28]. A topological boundary mode in a gapped two-dimensional SC must be present at the entire edge of such an MSH island as it traces the path where the Chern number changes. This is different for antiferromagnetic MSH-systems, where the presence of edge modes, that depend on the specific crystallographic direction of a boundary, has been identified as a hallmark for TNPSC[12,18,20,21]. From experiments alone it is challenging to identify the origin of boundary modes, even if the magnetic state has been determined, and often a comparison to a theoretical model is required to obtain a deeper understanding.

For future applications a controlled tuning of topological boundary modes is mandatory. In a recent experimental work, where a magnetic spin spiral was proximitized to an s-wave superconductor, it was shown that the dispersion of the edge modes sensitively depends on the actual magnetization direction at the boundary of the MSH system[21]. This observation shows a possible avenue towards the tuning of the properties of edge modes in 2D topological superconductors. A more versatile route to increased controllability could be achieved by interfacing two different topological superconducting phases, whose properties could be tuned ideally via electrical means. The observation of such edge modes at the boundary between two different topological superconducting phases has so far not been reported.

Here we report on the combined experimental and theoretical investigation of the magnetic and superconducting properties of hybrid MSH systems consisting of a Mn monolayer (ML) and a Mn bilayer (BL) on a superconducting Ta(110) surface. Low-temperature spin-polarized scanning tunneling microscopy (SP-STM) investigations reveal the presence of local AFM order in both Mn layers. Furthermore, highly spin-polarized low-energy edge modes are observed between the two different MSH regions. Such a spin-polarization of edge modes has not been reported before. Tight-binding model calculations allow us to explain the observed edge modes at the Mn-ML/Mn-BL boundaries as the result of the presence of two different TNPSC phases in the two MSH regions. These calculations reveal, that edge modes in TNPSCs can even emerge between two MSH systems, which differ only slightly in their electronic structure. Using the complex band structure we show that the spin-polarization of the edge mode at the boundary between two antiferromagnetic MSH systems is a general effect originating from a different decay of the edge mode into the bulk electronic structure of the adjacent TNPSCs.

## Results

### Experimental results on the Mn ML and Mn BL system

We have prepared ultra-thin films of Mn on Ta(110), see "Methods", and find that both the Mn ML as well as the Mn BL grow pseudomorphically. For a coverage of ~1.25 atomic layers grown at elevated temperatures we observe an almost fully closed ML with BL regions

attached to the step edges, see constant-current STM image and side view sketch in Fig. 1a, b, respectively.

Tunnel spectroscopy was performed at different positions of this sample as indicated by crosses in Fig. 1a, including the bare Ta substrate (i.e., a hole in the Mn ML), the Mn ML and the Mn BL, see Fig. 1b. Due to thermal broadening at the measurement temperature of 1.3 K, the density of states in the superconducting gap of the Ta does not go down to zero but is seen as a dip in the differential conductance (dI/dU) signal around zero energy. We observe similar spectra for the two different Mn layers with a slight increase of the dI/dU signal around zero-bias compared to the Ta substrate, demonstrating that due to the proximity to the substrate also the Mn layers are superconducting. The less pronounced superconducting gap is a manifestation of additional states inside the gap, which is a hallmark of magnetic moments in proximity to a superconductor[4].

First, we focus on the characterization of the Mn ML and prepare Mn ML islands by depositing sub-monolayer amounts at room temperature, see constant-current STM image in Fig. 2a. The islands are elongated along [001] and several straight edges along that direction, together with some well-developed edges along ⟨111⟩, can be seen. We have investigated the magnetic state of these Mn ML islands at low temperature using SP-STM, see current map in Fig. 2b (see "Methods"). The lines along [001] visible on the Mn ML island are separated by a distance equivalent to two atomic rows, demonstrating that the magnetic state is antiferromagnetic[29], see inset. (For further information on the magnetic state see Supplementary Notes 1 and 2).

Zero-bias dI/dU maps of the antiferromagnetic Mn ML islands show an increased signal along their edges, see Fig. 2c. This demonstrates an increased density of states in the center of the superconducting gap at the edges compared to the interior of the island. The intensity of this edge mode varies depending on the direction of the edges, as seen in a series of spectra taken across two different edges of this island: Figure 2d, e show the dI/dU signal (color-coded) as a function of the bias voltage and the path across the [1$\bar{1}$1] and [001] edges indicated in Fig. 2a, respectively. While we do observe an increase of the intensity within the gap for both edges, the spectra at the [001] edge (Fig. 2e) clearly feature a peak close to zero bias. For additional data with a different tip comparing the three high-symmetry edges, see Supplementary Note 4).

An unambiguous identification of the origin of such features only based on experiments is challenging. We rule out the formation of trivial YSR states only at the edges, as recently reported for van-der-Waals heterostructures[25,26], because our system shows strong coupling of magnetism and superconductivity also in the interior of the island. While we cannot rule out slight changes of the atom position or magnetic moments at the edge, this is not expected to lead to such a strong zero-bias intensity in our metallic system. Furthermore, we have no reason to believe that other trivial states, such as Andreev bound states (ABS), may be the origin of the features close to zero bias observed in our experiments. In principle, due to the presence of the boundary an electronic state may exist and could couple to the superconductor, giving rise to an ABS. However, due to the strong coupling in our metallic system such an ABS would merge with the coherence peaks[28]. Instead, antiferromagnetic MSHs are known to typically exhibit TNPSC, as also seen for a similar system, the antiferromagnetic Mn ML on Nb(110)[18], and we also observe the distinct edge mode properties at different edges, which are typical for TNPSC phases of this surface symmetry. Therefore, our experimental finding of an increased dI/dU signal near zero bias is in agreement with an interpretation of TNPSC with boundary modes to the surrounding trivial SC phase of Ta.

Next, we investigate the properties of the Mn BL, and the boundary between the ML and BL, using a sample with a nearly closed ML and BL islands. We find that the BL islands exhibit rather long straight edges along [001], see current map in Fig. 3a (and

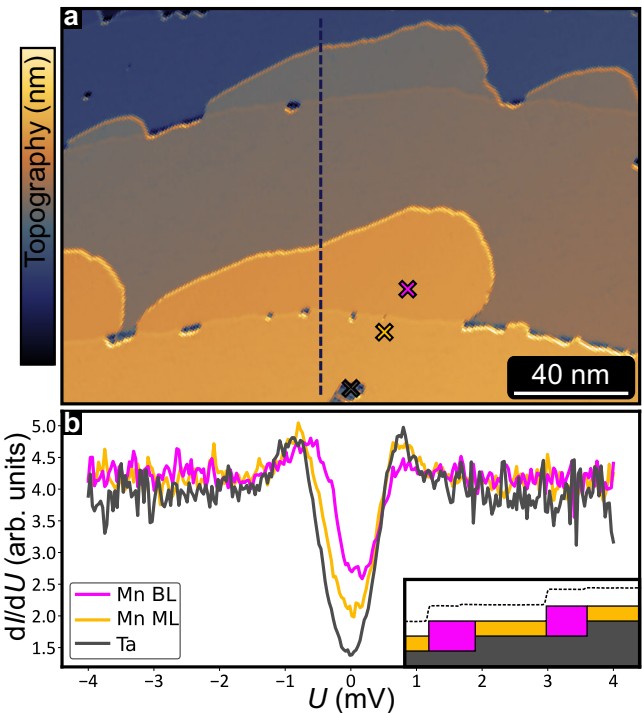

**Fig. 1 | Superconducting properties of Mn/Ta(110). a** Partially differentiated constant-current STM image of a sample of 1.25 atomic layers of Mn on Ta(110). **b** Tunnel spectra acquired at 1.3 K on the bare Ta, the Mn ML and the Mn BL, at the positions indicated by colored crosses in a. Inset shows line profile and side view sketch of the sample along the dashed black line in a. Measurement parameters: a: $U = + 60$ mV, $I = 1$ nA.; b: stabilization bias $U_s = + 4$ mV, stabilization current $I_s = 1$ nA, modulation bias $U_{mod} = 50$ $\mu$V, $T = 1.3$ K; all: Cr-bulk tip.

corresponding constant-current STM image in Supplementary Fig. S5). SP-STM reveals that the BL is also antiferromagnetic, see Figs. 3a, b, and here both the ML and the BL have all spins aligned out-of-plane with a strict phase relation between them due to antiferromagnetic inter-layer coupling, see sketch in Fig. 3c (and also Supplementary Fig. S1, and S2). The termination of the antiferromagnetic state of the BL at the [001] edges results in an edge with parallel spins. The two opposite magnetization directions at the edge can be identified in the SP-STM current image in Fig. 3a by two distinct signal intensities, see green and red markers. The edge indicated by the left rectangle exhibits two single atomic kinks resulting in opposite magnetization directions for adjacent segments of this island edge. Similar to the case of the Mn ML we observe edge states between the Mn BL and the trivial Ta substrate, see Supplementary Fig. S9. However, due to the sample morphology only the [001] edges of the elongated BL islands to the Ta substrate could be characterized, while the [1$\bar{1}$0], and [1$\bar{1}$1] edges are inaccessible. Nevertheless, based on the same arguments as put forward regarding the interpretation of the ML edge modes, our experiments suggest that also the Mn BL hosts a TNPSC phase with boundary modes to the trivial SC of the Ta substrate.

## The boundary between Mn ML and Mn BL

We have seen that both antiferromagnetic layers, the Mn ML and the Mn BL, show a superconducting gap and develop edge modes at the boundary to the trivial superconducting phase of the Ta substrate. This presents an opportunity to investigate the transition between the two different antiferromagnetic MSH systems. For the sample area indicated by the left rectangle in Fig. 3a we have measured the zero-bias d$I$/d$U$ intensity, see Fig. 4a. We observe an increased zero-bias signal at the [001] edges between the Mn BL and the Mn ML, indicating that there is an edge mode between the two regions. We find that this edge

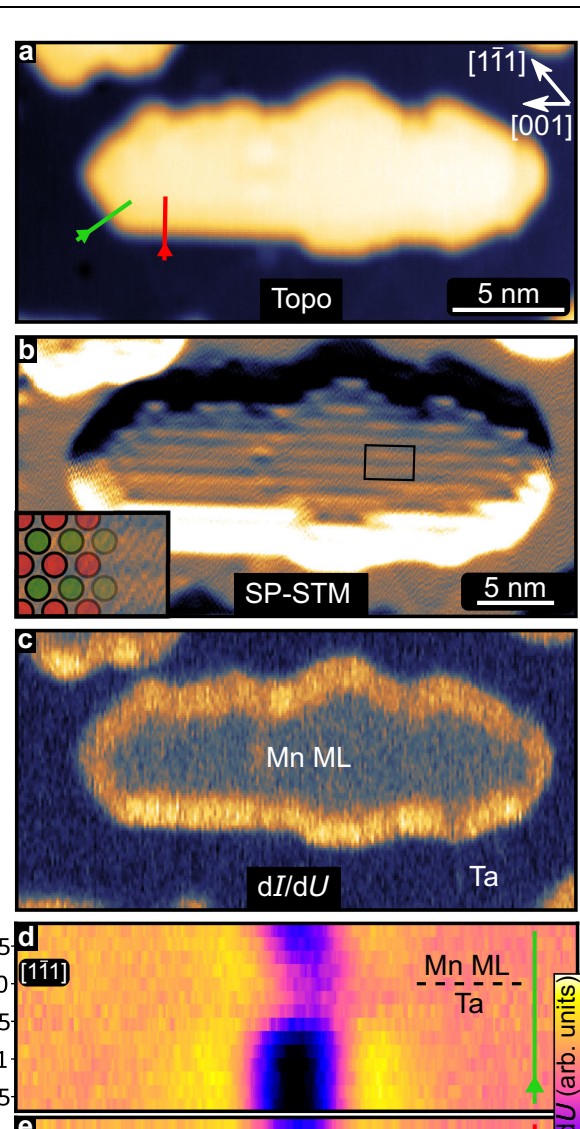

**Fig. 2 | Magnetic State and Edge Mode of the Mn ML on Ta(110). a** STM constant-current image showing an island of the Mn ML with several straight edges. **b** SP-STM current map of the same island using a spin-polarized tip, inset shows the row-wise antiferromagnetic order; fast scan direction is vertical; color range $1 \pm 0.15$ nA. **c** Zero-bias d$I$/d$U$ map of the same island (multi-pass mode); color range $1.00 - 5.63$ arb. units. **d**, **e** Spatially- and energy-resolved d$I$/d$U$ signal across a [1$\bar{1}$1]/[001] edge along the green/red line marked in (**a**) respectively. Each spectrum was normalized; color range $0.45 - 1.20$ arb. units, for raw data see Supplementary Fig. S4. Measurement parameters: a: $U = + 4$ mV, $I = 1$ nA; b: $U = + 5$ mV, $I = 1$ nA; c: $U = 0$ mV, $U_{mod} = 50$ $\mu$V, $T = 4.2$ K, measured at the tip-sample distance obtained in a; d,e: spectra with stabilization bias $U_s = + 4$ mV, stabilization current $I_s = 1$ nA, modulation bias $U_{mod} = 50$ $\mu$V; **a**, **b**, **d**, **e**: $T = 1.3$ K, all: Cr-bulk tip.

state vanishes when the superconductivity is quenched by an external magnetic field (see Supplementary Note 6), verifying that it originates from the superconducting state. Moreover, we observe two distinct intensities for this BL/ML edge mode: as seen in Fig. 4a the middle segment of the edge has a lower intensity than the ones on the left and

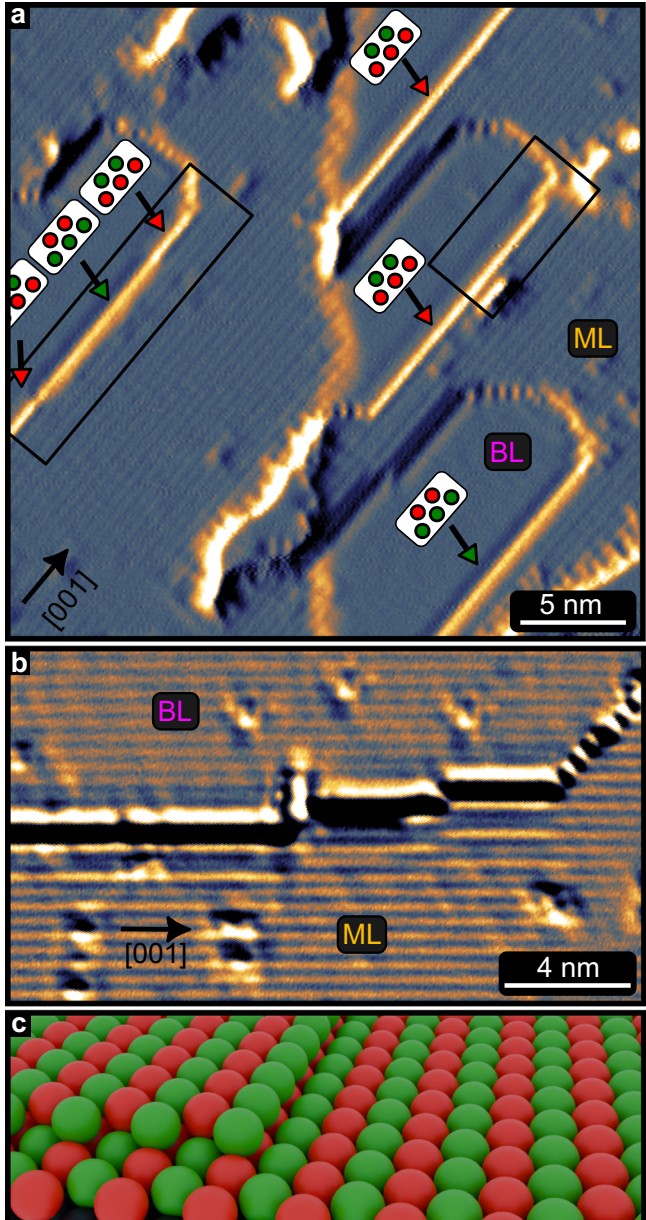

**Fig. 3 | Magnetic State of the Mn ML and BL on Ta(110). a** SP-STM current map of Mn BL islands on a nearly closed Mn ML (note the buried Ta step edge in the center of the image); red/green markers indicate the two different contrast levels; color range 0.705 − 1.475 nA. **b** SP-STM current map of a Mn BL island (top), next to an area of Mn ML (bottom). Both areas show the same magnetic stripe contrast indicative of antiferromagnetic order. Fast scan axis is vertical; color range -2 ± 0.15 nA. **c** Sketch of a region of Mn BL and ML. The top layer atoms sit in the fourfold hollow site positions. Measurement parameters: a: $U = +5$ mV, $I = 1$ nA, $T = 1.3$ K; b: $U = −15$ mV, $I = 2$ nA, $T = 4.2$ K; all: Cr-bulk tip.

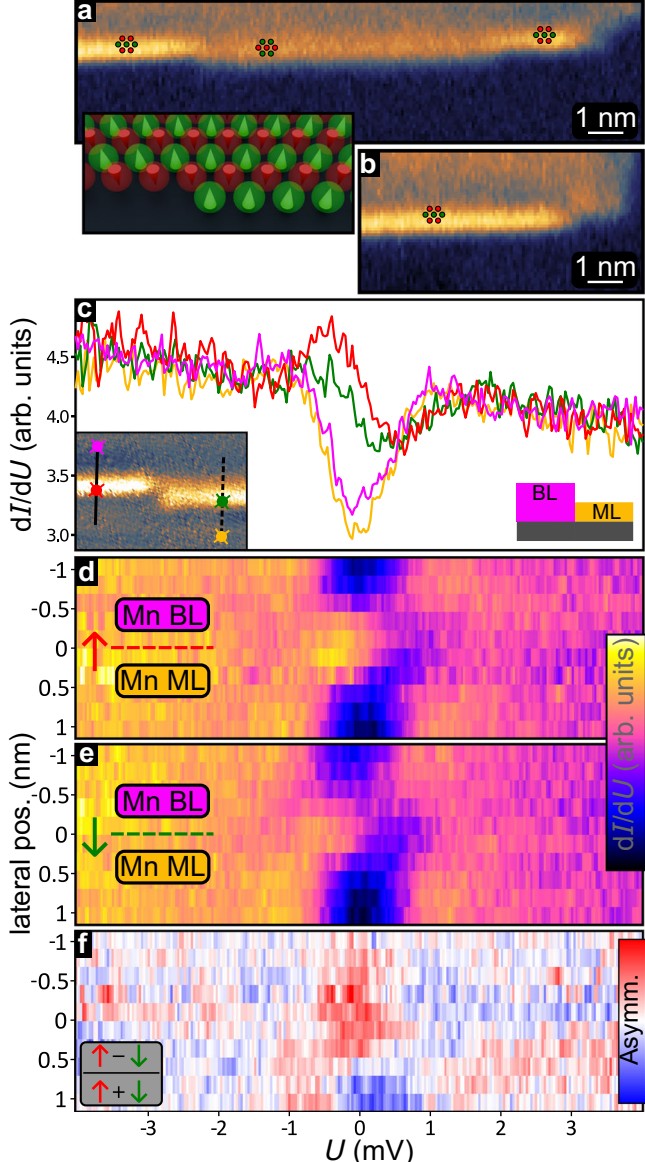

**Fig. 4 | Spin-polarized edge mode between ML and BL. a, b** Zero-bias d$I$/d$U$ maps at the sample areas indicated by black rectangles in Fig. 3a (constant-height mode); red/green markers show the magnetization of the different segments of the [001] edge between ML and BL (to scale); inset sketches the edge magnetization across an atomic kink; color range -0.58 − 6.40 arb. units. **c** Spectra of the ML, the BL, and the oppositely magnetized [001] edges; right inset shows a side view sketch of the sample, left insets shows the positions of the spectra in an SP-STM current map. **d, e** Spatially-, spin-, and energy-resolved d$I$/d$U$ signal across [001] edges with opposite magnetization directions, see lines in inset to **c**; Each spectrum was normalized, for raw data see Supplementary Fig. S7; color range 0.7 − 1.20 arb. units. **f** Calculated asymmetry of the data shown in d,e. Measurement parameters: **a, b**: $U = 0$ mV, $U_{mod} = 50 \mu$V, $T = 1.3$ K, measured with the tip moving on a plane parallel to (110); **c–e** stabilization bias $U_s = +4$ mV, stabilization current $I_s = 1$ nA, modulation bias $U_{mod} = 50 \mu$V. Left inset in a: $U = +5$ mV, $I = 1$ nA; all: Cr-bulk tip, $T = 1.3$ K.

right, which in turn have the same intensity as the edge shown in Fig. 4b (see right rectangle in Fig. 3a for its position and edge magnetization). This two-stage contrast at zero-bias is correlated to the magnetization direction of the respective edges, compare Fig. 3a and Fig. 4a, b. We therefore conclude that our antiferromagnetic Mn film exhibits an edge mode at the BL/ML boundary, which shows a spin-polarization that is governed by the magnetization direction at the specific edge.

In order to characterize this spin-polarized low-energy edge mode between Mn ML and Mn BL we perform spatially-, spin-, and energy-resolved d$I$/d$U$ spectroscopy across two [001] edges with opposite

spin orientations. Spectra taken on the two magnetically distinct [001] edges are shown in Fig. 4c, together with the spectra on the adjacent Mn ML and Mn BL, see inset. The spectra of both edges exhibit a peak-dip structure in the superconducting gap, but their intensity differs due to their different spin orientations. These spectra are part of a series of equidistant spectra across each edge, and the full datasets are displayed in Fig. 4d, e. Indeed, when moving from the BL (top) to the ML (bottom) we find that the d$I$/d$U$ intensity in the superconducting

**a**

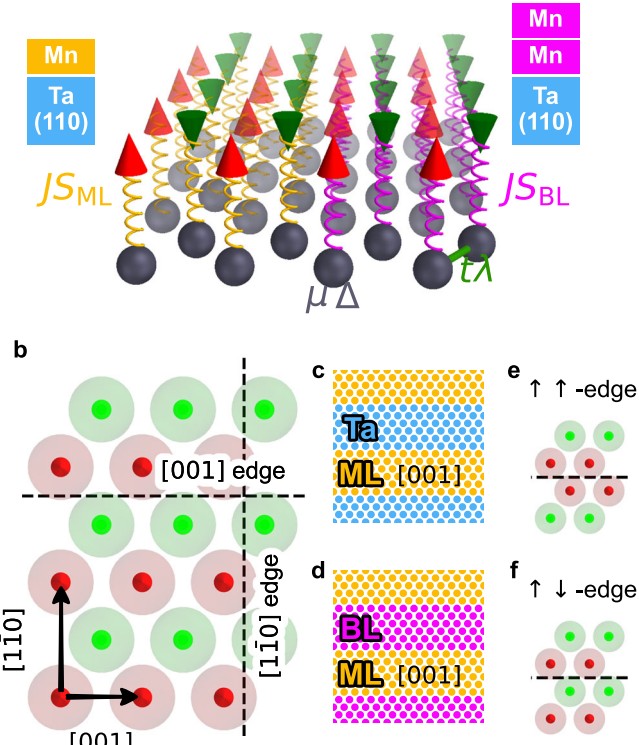

**b**

**Fig. 5 | Computational setup for the tight-binding model. a** Sketch of a magnetic layer containing an AFM state (cones) and a superconducting layer (spheres). Red and green cones represent the directions of the magnetic moments. In the model the magnetic and superconducting properties are mapped onto a single atomic layer. The onsite parameters $\mu$, $\Delta$, which reflect the chemical potential and the superconducting order parameter, and the hopping parameter $t$ as well as the Rashba-type SOC parameter $\lambda$ are marked. The coupling between the magnetic and the superconducting layer is indicated by springs between the respective lattice sites. The coupling strength is denoted as $JS_{ML}$ for the Mn/Ta(110) system and as $JS_{BL}$ for Mn/Mn/Ta(110). The spatial transition from a ML system to a BL system is represented by a change in the local coupling, as marked by the springs of different colors. **b** Unit cell of the AFM state in the magnetic layer. Edges in two crystallographic directions are marked. **c** Stripe geometry for a ML stripe interfacing a trivial Ta domain. The stripes are periodic in [001] direction and alternating in [1$\bar{1}$0] direction. Each ML or Ta stripe has a width of 200 unit cells in the tight-binding calculation. **d** Sketch of a similar setup as in panel **c**, but here ML and BL stripes are alternating in the [1$\bar{1}$0]-direction. **e** ↑↑-edge in [001] direction. **f** ↑↓-edge in [001] direction.

gap is significantly increased at the edges, with a clear maximum just below zero-bias and different signal strengths for the two oppositely magnetized edges. The difference due to the spin-polarization can more easily be seen in the asymmetry map displayed in Fig. 4f, where the difference of ↑ (Fig. 4d) and ↓ (Fig. 4e) divided by their sum is plotted as a function of position and energy. Whereas edge states have been observed before in two-dimensional MSH systems, they were located at the boundary between the magnetic TNPSC and the non-magnetic SC substrate[18,21]. The existence of an edge mode at the boundary between what appears to be two different magnetic TNPSCs –the antiferromagnetic Mn ML and BL in our case– raises questions about how the formation of such edge modes can be understood.

## Tight-binding model

The experiments demonstrate a low-energy spin-polarized edge mode between two different antiferromagnetic MHSs, represented by a Mn ML and a Mn BL on a Ta(110) surface. To understand the origin of this edge mode and its spin-dependent properties we use a tight-binding

model. In the model we map all properties onto a single layer, whose superconductivity is influenced by the magnetization of the ML or BL via a coupling $J$ to the spins $S$ (Fig. 5a). Further model parameters are the chemical potential $\mu$, the superconducting order parameter $\Delta$, the hopping term $t$, and the Rashba-type spin-orbit coupling (SOC) term $\lambda$. The Hamiltonian is given by

$$
\begin{aligned}
H = & -\mu \sum_{i,\sigma} c_{i,\sigma}^\dagger c_{i,\sigma} + \Delta \sum_i (c_{i,\uparrow}^\dagger c_{i,\downarrow}^\dagger + c_{i,\downarrow} c_{i,\uparrow}) \\
& + t \sum_{i,j,\sigma} c_{i,\sigma}^\dagger c_{j,\sigma} - i\lambda \sum_{i,j,\sigma,\sigma'} c_{i,\sigma}^\dagger (\hat{e}_{r_i - r_j} \times \boldsymbol{\sigma}) c_{j,\sigma'} \\
& + JS \sum_{i,\sigma,\sigma'} c_{i,\sigma}^\dagger (\hat{s}_i \cdot \boldsymbol{\sigma}) c_{i,\sigma'}
\end{aligned}
\tag{1}
$$

where $c_{i,\sigma}^\dagger$ and $c_{i,\sigma}$ are the electronic creation and annihilation operators at the position $r_i$ with spin $\sigma$, $\hat{e}_{r_i - r_j}$ is the normalized connection vector between lattice sites $i$ and $j$, and $\boldsymbol{\sigma}$ is the vector of the Pauli spin matrices. The geometry of the bcc(110) surface and the anti-ferromagnetic ground state of the Mn ML and BL are displayed in Fig. 5b.

Previous theoretical investigations have demonstrated that AFM order can in general lead to TNPSC[18,21], and we obtain the same for a large region of the parameter space (see Supplementary Note 11 and Supplementary Fig. S20). While our findings are independent of the exact choice of parameters, as a starting point we use parameters that lead to a different number of NPs for the ML and the BL, as this is a tentative explanation for the experimental observation of a boundary mode between them (see "Methods section" for a detailed discussion of the values). To model boundaries along the high-symmetry crystallographic directions between different surfaces we use a stripe geometry, where each stripe represents one material and is infinite in one direction, whereas the properties of the stripes alternate in the other direction. This ribbon geometry is sketched in Fig. 5c for an edge along the [001] direction (cf. Fig. 5b) between the Mn ML and Ta. In the calculations each stripe has a width of 200 unit cells. For boundaries along [1$\bar{1}$0] the stripes are oriented accordingly. To model a ML/BL boundary we use a stripe geometry with alternating BL and ML domains (Fig. 5d). In this case two different spin configurations at the boundary are possible: either the two layers terminate with the same spin direction (Fig. 5e), which leads to an ↑↑ edge at the boundary as in the case where the top layer of the BL is considered (see sketch in Fig. 3c); or the boundary continues the antiferromagnetic order resulting in a ↑↓ edge (Fig. 5f), which is realized in the lower Mn layer of the BL (see sketch in Fig. 3c).

## Band structures and edge modes of MSH systems

First, we characterize the bulk properties of the ML and the BL by calculating their respective band structures (Fig. 6a, b). The ML exhibits six nodal points at the Fermi energy (red sheet) which are highlighted by yellow points. Two pairs of nodal points are located on the boundary of the two-dimensional Brillouin zone (2D-BZ) and one pair is located in the center (Fig. 6a). The BL has only two pairs of nodal points located on the boundary of the BZ (Fig. 6b). To identify whether these nodal points are topological we investigate boundaries from these TNPSC to the trivial Ta surface, where the occurrence of edge modes is a hallmark of topologically distinct properties. The bulk band structures of ML and BL, projected along the high symmetry direction of the 2D-BZ denoted by $k_{[001]}$, are shown in black in Fig. 6c, e. The red points mark the additional states arising for a stripe geometry along [001] with a boundary of the ML or BL, respectively, to the trivial Ta surface (cf. Fig. 5c). For both systems, the ML and the BL, an edge state (red line) connects the two nodal points located on the boundary of the BZ, in agreement with the experimental observation of low energy modes at the boundaries between each layer and the Ta substrate. Similar stripe geometry calculations have been performed for the

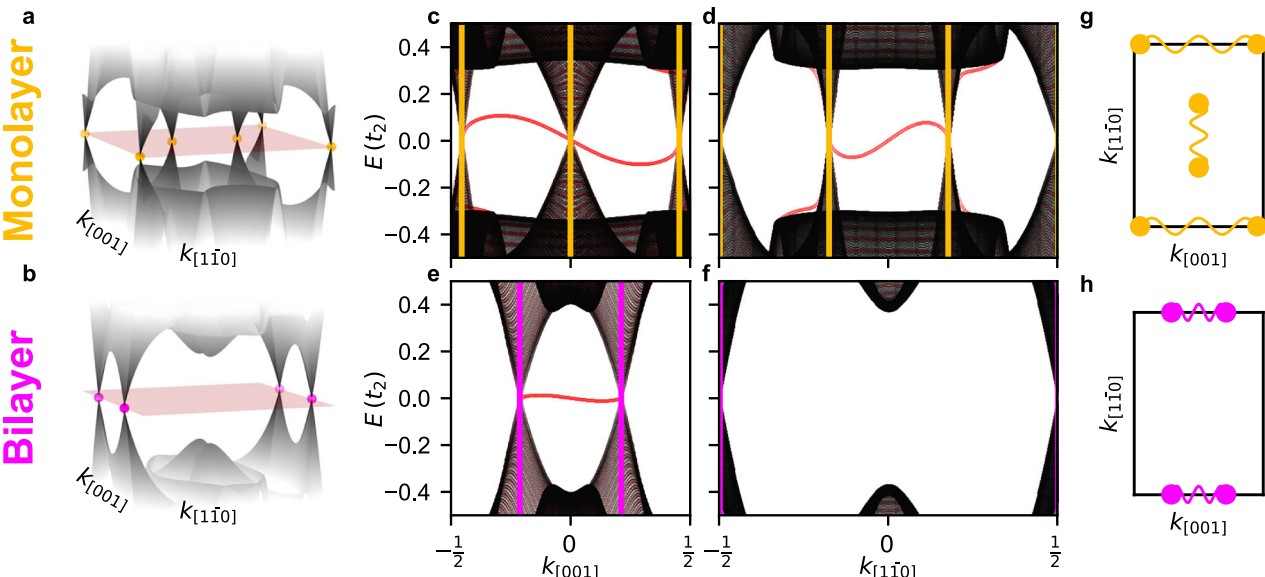

**Fig. 6 | Band structure calculations for monolayer and bilayer. a, b** Energy dispersion of the 2D-BZ around the Fermi energy for a Mn monolayer (**a**) and a Mn bilayer (**b**). The red surface represents the Fermi energy and the colored round markers highlight the positions of the nodal points. **c–f** Projected band structures from an infinite sized sample (black) and the dispersion along the periodic direction of a stripe geometry (red). The dispersions are shown for the ML (**c, d**) and the BL (**e, f**). In both cases the magnetic domains are adjacent to Ta domains as sketched in

Fig. 5c. **c, e** Band structure projected onto the [001] direction of the 2D-BZ. The ribbon has an infinite length in the [001] direction and a width of 200 magnetic unit cells in the [1$\bar{1}$0] direction. **d, f** Band structure projected onto the [1$\bar{1}$0] direction in which the ribbon is periodic in the same direction. The positions of the bulk nodal points from panels (**a, b**) are marked with vertical yellow (ML) and purple (BL) lines. **g, h** 2D-BZ with the positions of the nodal points marked. Nodal points, which are connected by edge states in the ribbon geometry are connected by curved lines.

[1$\bar{1}$0]-direction and the corresponding projected band structures for the ML and BL are shown in Fig. 6d, f, respectively. In this crystallographic direction only the ML has an edge mode, which connects the two nodal points located in the middle of the 2D-BZ. The pairwise connection of nodal points by edge modes seen in Fig. 6c–f is summarized in Fig. 6g, h for the ML and BL, respectively. As all nodal points form pairs, connected by edge modes, we conclude that all nodal points are topological.

To investigate the boundary between the two different TNPSCs we create a pattern of alternating stripes of the ML and the BL with the stripes infinitely extended in the [001]-direction (Fig. 5d). To model the antiferromagnetically coupled Mn layers (cf. Fig. 3c), we assume the ↑↑-edge (cf. Fig. 5e); note that similar results are found for a ↑↓-edge (see Supplementary Fig. S10). In the band structure along the [001] direction (Fig. 7a) the positions of the nodal points obtained from the ML and BL bulk calculations (cf. Fig. 6) are marked by vertical yellow (ML) and purple (BL) lines. Due to the bulk-like properties in the wide stripes the same nodal points are also present in the stripe geometry. Two modes emerge around zero energy between nodal points (Fig. 7a). In contrast to the edge modes at an interface with a trivial phase (Fig. 6) these edge modes connect nodal points of the two different TNPSC systems, i.e., a ML nodal point is connected to a BL nodal point. This is qualitatively different to previous work on edge modes between TNPSC and trivial states[12,18,21]. Here we find edge states emerging between two different TNPSCs, which differ only in the coupling strength to the magnetic layer.

The spectral functions obtained in our tight-binding calculation for the four rows of atoms (two in the ML and two in the BL) located closest to the boundary are displayed in Fig. 7b, c for the spin-up (↑) and spin-down (↓) states, respectively. First, these graphs demonstrate that indeed these modes connecting NPs are located at the ML/BL boundary, and second, it can be seen that the ↑ and ↓ channels differ in their contribution to the edge mode (Fig. 7b,c), resulting in a finite spin-polarization. Some states outside of the superconducting gap also show a spin polarization. A spin polarization can also be seen for an edge along [1$\bar{1}$0] (see Supplementary Fig. S11).

In the experiments the BL/ML edge mode manifests as an increased d$I$/d$U$ signal at zero-bias compared to the bulk of the MSHs (Fig. 8a, b). The ↑-edge and the ↓-edge terminate with an opposite magnetic moment, and using the same magnetic tip the two edges have a different signal intensity, demonstrating that the edge mode is spin-polarized. The line profiles across the edges and the position-dependent spin polarization SP = $\frac{\uparrow - \downarrow}{\uparrow + \downarrow}$ are shown in Fig. 8c, d, respectively. To compare the experimental findings with the calculated edge states (cf. Fig. 7a) we have performed SP-STM simulations (Fig. 8f, g) based on the spin-polarized generalization of the Tersoff-Hamann model[30,31]. In this model, the tunneling current is proportional to the spin-dependent local density of states (LDOS) at the position of the STM tip, i.e., in the vacuum (see "Methods" for details on the STM simulation).

Using an ↑↑-edge (Figs. 5e and 7) we perform simulations for two opposite tip magnetization directions, which is equivalent to the case of two opposite edges measured with the same magnetic tip as in the experimental setup (Fig. 8a, b). For simplicity, we assume an ideal tip with a 100% spin polarization. In this case, the simulated SP-STM images for the two tip magnetizations (Fig. 8f, g) correspond to the vacuum LDOS of the spin-up (↑) and spin-down (↓) channel (for simulations with a 50% tip spin-polarization see Supplementary Fig. S12 and Supplementary Note 8). Comparison shows that the signal near the boundary is strongly spin-polarized, with the maximum intensity in the ↑ channel.

The line profiles of the two spin channels (Fig. 8h) show that the spin-up and the spin-averaged signal have their maximum intensity on the ML side. The obtained spin polarization (Fig. 8i) also exhibits its maximum value on the ML side of the interface. When moving further into the ML the spin polarization is oscillating, while it is going to zero comparably fast in the BL. This indicates a different decay of the edge state into both MSH systems. Given the much smaller spin polarization of the experimental tip we find good qualitative agreement between the experimental and theoretical data, both showing an edge mode with a sizable spin-polarization.

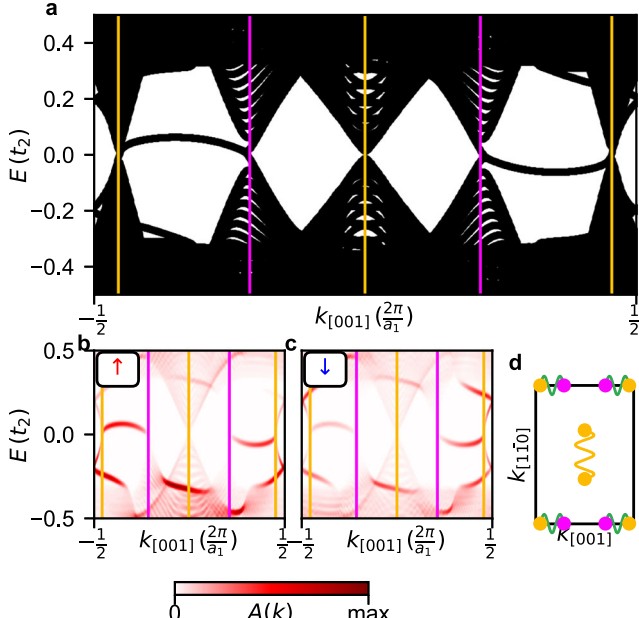

**Fig. 7 | Band structure for the ML/BL stripe geometry with an ↑↑ edge. a** Band structure along $k_{[001]}$ for ribbons, which are periodic in this direction. The positions of the nodal points from the ML bulk and the BL bulk are marked by vertical lines (cf. Fig. 6). **b**, **c** Spectral functions of the first two rows of the ML and the first two rows of the BL counting from the boundary. The spectral function is shown for the electron spin-up orbitals (**b**) and the electron spin-down orbitals (**c**). **d** Sketches of the 2D-BZ with the positions of the bulk nodal points of the ML and BL. Nodal points which are connected by edge modes for the [001] or [1$\bar{1}$0] geometry are indicated by wavy lines. For a ML/BL interface with an ↑↓ edge and a stripe geometry in the [1$\bar{1}$0]-direction see Supplementary Figs. S10 and S11.

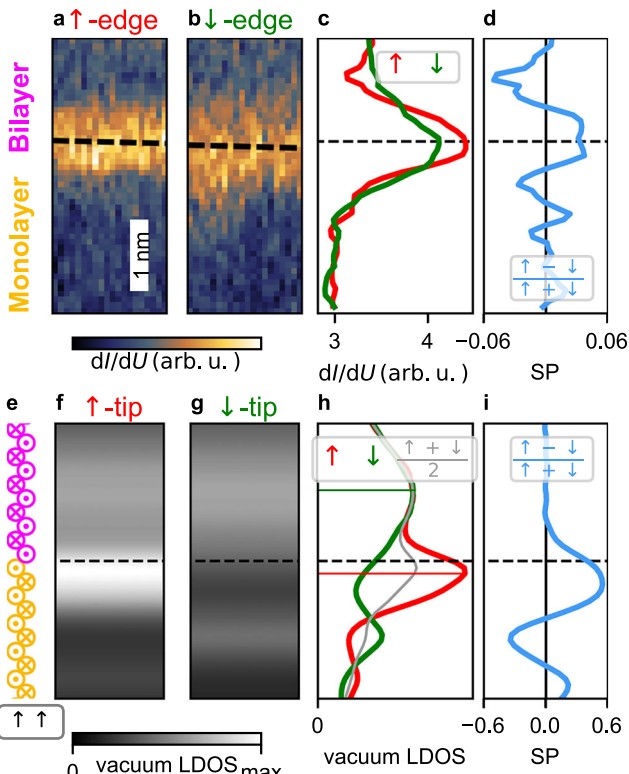

**Fig. 8 | Measured and simulated zero-bias SP-STM images for the ML/BL boundary with an ↑↑ edge. a**, **b** Zero bias d$I$/d$U$ maps of an ↑-edge and a ↓-edge between the BL(top) and ML(bottom) measured by retracing the topography measured at 4 mV. The black dashed lines shows the estimated edge position, see Supplementary Fig. S7c for more information. **c** d$I$/d$U$ line profiles across the two edges in a,b created by averaging ten scan lines. **d** Calculated asymmetry between the two plots in (**c**). **e** Sketch of the magnetic moment for a ↑↑ edge in the [001]-direction between the ML and BL. **f**, **g** Simulated zero-bias SP-STM contrast for a 100% spin polarized tip, which is ↑ polarized (**f**) or ↓ polarized (**g**).The boundary between both domains is displayed as a black dashed line. The tip height is 4 Å and the averaged voltage interval is [ − 0.02$t_2$, 0.02$t_2$]. **h** Integrated SP-STM signal along [001] for spin-up (red), spin-down (green) and spin-averaged (gray). The position of the maximum intensity is marked by a thin solid line. **i** Spin polarization of the SP-STM signal calculated from the values shown in (**h**).

## Spin-polarization of MSH edge modes

Based on the tight-binding model we can explain the existence of an edge state between two different TNPSC and its spin-polarization, and found good qualitative agreement with the experimental observations. However, the question arises how general the result of a spin-polarized edge state is, i.e., does it occur also for different boundary terminations and other model parameters. In correspondence with the experiment, we have so far discussed a boundary with an ↑↑-edge (Fig. 5e). However, in general an ↑↓-edge can also occur (Fig. 5f), e.g., for a buried step edge in the Ta; note that in the samples studied here the directions of the buried Ta step edges do not coincide with the [001] direction, preventing an experimental study of this boundary type. While the spin polarization at an ↑↑-edge seems rather intuitive, it is not obvious whether an edge mode at the ↑↓-edge can be spin-polarized as well (For a boundary between ML or BL and the Ta surface see Supplementary Fig. S13).

To resolve this issue, we have performed tight-binding calculations for a ML/BL boundary with an ↑↓-edge (Fig. 9a). As seen from the spin-polarization of the vacuum LDOS (Fig. 9b) and the position dependent spin-averaged LDOS at the atoms (Fig. 9c) there is a spin-polarized edge state also at this boundary (for band structure see Supplementary Fig. S10 and for SP-STM simulations see Supplementary Fig. S14). Interestingly, the spin polarization of the LDOS is very similar for the ↑↓-edge (Fig. 9b) and the ↑↑-edge (Fig. 8i) and both curves show the characteristic shift towards the ML. The edge state decays exponentially in both directions and in addition displays oscillations in the ML (Fig. 9c), which leads also to an enhanced oscillation of the spin polarization in the ML (Fig. 9b). The effective decay constant is obtained from a fit to the spin-averaged LDOS of atom sites far from the boundary (starting from the outermost atoms

from the interface visible in Fig. 9c) and it is larger in the ML ($\kappa_{ML}^{eff} = 0.37$) than in the BL ($\kappa_{BL}^{eff} = 0.25$) (cf. Supplementary Fig. S15 and Supplementary Note 9).

The spin-resolved LDOS displayed in Fig. 9d–g for the atom sites closest to the edge (denoted according to the sketch in Fig. 9a) demonstrates that the edge mode is highly spin-polarized in the direction of the respective magnetic moment. When comparing the different spin channels for these four lattice sites one can see that the ↑ channel dominates, leading to a net spin polarization in the LDOS. Both spin contributions can compensate on an atomic scale in the BL as the edge state is only slowly decaying (Fig. 9b). Due to the faster decay of the edge state into the ML, the spin termination on the ML side defines the spin polarization of the edge mode and the maximum spin polarization is shifted into the ML (Fig. 9b).

In the case of surface states it is known that their decay into the bulk is an intrinsic material property and can be calculated via the complex band structure[32]. In our system that corresponds to the decay of the edge mode into the two adjacent antiferromagnetic MSH films. The decay constants are calculated individually for the ML and the BL as they stem from the respective bulk band structures (for details see the "Methods section"). Figure 9h shows the decay constants

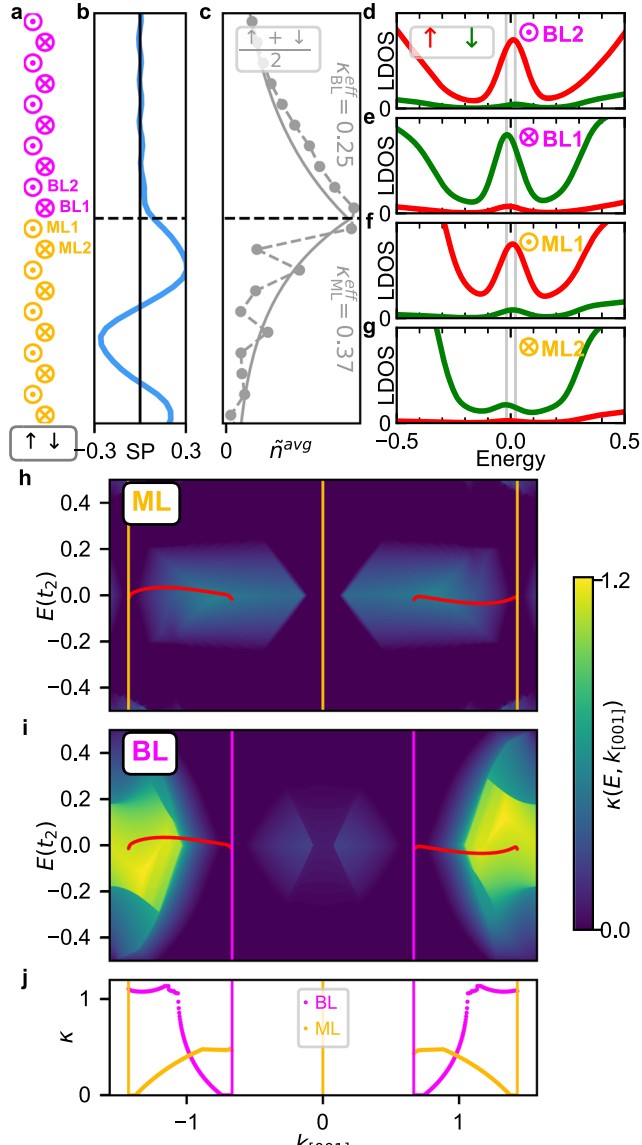

**Fig. 9 | LDOS for the ↑↓-edge and decay constants for surface states calculated from the complex band structure. a** Sketch of the spin structure close to the edge. **b** Spin polarization for the ↑↓-edge, calculated from the vacuum LDOS at 4 Å. **c** spin-averaged LDOS for lattice sites close to the boundary. The LDOS is integrated in an energy interval $E \in [-0.02t_2, 0.02t_2]$ around the Fermi energy. The solid line represents a fit, where the fitted decay values $\kappa_{BL}^{eff}$ and $\kappa_{ML}^{eff}$ are noted. The dashed line is a guide to the eye. **d–g** LDOS for the spin-up (red) and spin-down (green) orbitals for the lattice sites BL2, BL1, ML1, and ML2, respectively, as defined in (**a**). The energy window used for the integration in (**c**) is marked by vertical gray lines. **h, i** Decay constant $\kappa\left(E, k_{[001]}\right)$ as a function of the energy $E$ and the wave vector $k_{[001]}$ of a surface state. **h** Decay constants for the ML and (**i**) for the BL. The edge mode from the ML/BL boundary (cf. Fig. 7a) is marked in red. The positions of the bulk nodal points of the ML and BL (cf. Fig. 6) are indicated by purple and yellow lines. **j** Decay constants for the edge modes from (**h, i**).

$\kappa_{ML}(E, k_{[001]})$ for the ML, which are universal for all possible boundaries. The edge mode at the ↑↓-edge is marked in red and the positions of the bulk nodal points are indicated by vertical yellow lines. The areas of vanishing $\kappa$ (dark blue) coincide with the bulk band structure (cf. Fig. 6c) and indicate an infinite decay length. Finite values for $\kappa$ represent states which lie in the projected bulk band gap, resulting in a faster exponential decay. For the ML, the states of the edge mode (red line) have a small $\kappa$ at the ML NPs, and it increases when moving

towards the position of the BL NPs (see Fig. 9j). The decay constants for the BL (Fig. 9i) show a similar correlation with the projected bulk band structure (cf. Fig. 6e). Also here the edge mode states have the smallest $\kappa$ near the respective NP, in this case the BL NP, and increase towards the NPs of the other MSH system (the ML), see Fig. 9j. This k-dependent decay of the edge mode can also be seen in the spectral function (Supplementary Fig. S16).

The total decay of the edge state into each MSH system is given by an integration of all $k_{[001]}$ contributions, weighted with the spectral function. While the explicit calculation of the decay length for the whole edge mode is quite tedious, an effective decay constant of the edge state is obtained by a fit as in Fig. 9c, which reveals an overall faster decay into the ML. Different decay lengths can be expected in general for boundaries between any two systems, due to their different electronic structures. Consequently, any edge mode between two TNPSCs is expected to have a net spin-polarization, independent of the details of the boundary and its specific termination. At an ↑↑-edge the spin polarization can be further enhanced by the parallel spin termination at the boundary.

## Discussion

The experimental observation of a spin-polarized edge mode at the boundary between the AFM Mn ML and the AFM Mn BL on a super-conductor can be explained by our tight-binding model, assuming topological nodal-point superconductivity with slightly different parameters in both regions. While we cannot experimentally prove the topological nature of edge modes at the boundary between these two AFM MSHs, we can clearly observe their spin-polarization, and find good agreement with the tight-binding model. The edge states connect NPs from the different TNPSCs, which are located at different positions on the BZ boundary.

To investigate how general this finding is we have varied the coupling strengths $JS$, thereby continuously decreasing the difference between the two neighboring MSH systems until they become identical (Supplementary Fig. S16, Supplementary Fig. S17, Supplementary Fig. S18, Supplementary Note 10). In this scenario, the NP structure of one MSH transitions into that of the other MSH. Interestingly, edge states emerge even for small differences in the coupling strengths and also for the same number of nodal points in both MSHs. The only requirement for an edge mode between two TNPSC is the presence of NPs in the projected band structure for both MSH systems, so that the different NPs can be connected for the specific direction of the boundary. Consequently, depending on the symmetry and the MSH properties, such boundary modes can arise in several directions or be confined to one direction only. The weight of the edge state in the LDOS, which is a measurable fingerprint, scales with the distance of the connected nodal points. The emergent different electronic structures of the two adjacent MSH systems directly lead to different decay lengths of the edge modes, which in turn governs their spin-polarization. Both the width of boundary modes and their spin-polarization impact the lateral transport properties of such MSH systems with TNPSC.

In contrast to gapped systems that have edge modes at boundaries between phases of different Chern number, our study reveals that systems of nodal-point superconductors allow for a tuning of the edge states. The distance between the nodal-points in adjacent TNPSC strongly varies with the parameter set and is directly related to the dispersion of the edge state. A lateral change in the band structure might also be achieved in a single TPNSC film by external means such as applying local strain or electric fields. For example, a gate could be fabricated which covers one part of the TNPSC. An applied voltage would then change the electronic structure below the gate with respect to the uncovered TNPSC, triggering the formation of a spin-polarized edge mode at the boundary between gated and non-gated parts of the MSH. The combination of two or more antiferromagnetic

layers with the same superconducting substrate, or a patterned device with electrical gates, represents a potential next step in the investigation of MSHs. The tailoring of boundaries between regions hosting different topological phases creates new possibilities for the manipulation of topological edge modes.

## Methods

### Sample preparation

The sample preparation and experiments were done in ultra-high vacuum. The Ta(110) single crystal surface was cleaned by heating to temperatures > 2300 K. This removes the oxygen reconstruction and reveals the unreconstructed (110) surface. For more details see ref. 33. Mn was evaporated from a PBN Knudsen cell at a temperature of 690 °C at a flux of ~0.2 atomic layers per minute. For samples including both the Mn ML and Mn BL, the Mn was deposited starting 8-10 min after the last flash annealing. This means the Ta crystal was still at elevated temperatures of approx. 100–200 °C during the deposition. For samples with sub-monolayer coverage the Mn was deposited after the Ta crystal had cooled to room temperature. The samples were then cooled down to 4.2 K or 1.3 K within the STM.

### STM measurements

The STM measurements were performed using a Cr bulk tip, etched in 1 M HCl solution. The tip was first cleaned via field emission and subsequently sharpened by voltage pulses and tip-sample contact. Even though Cr is antiferromagnetic the apex of our tip is not always magnetic as it can be covered by non-magnetic material. In that case gentle tip-surface collisions were performed to pick up Mn atoms to make the tip magnetic.

SP-STM measurements are performed in constant-current mode. The magnetic state of the ML and DL is shown using current maps (Figs. 2b and 3b) instead of topography images, as the former provide much better contrast when scanning across regions of different height, see for instance Supplementary Fig. S6a, b. Even when optimizing the magnetic contrast on a single layer by adjusting the color map, the current signal was generally superior. Note that at step edges the feedback artifacts in the fast scan direction are typically larger in the current maps compared to the topographic images.

For maps of differential tunnel conductance and tunnel spectroscopy both the current and voltage offsets were adjusted until the current was zero at zero bias (or very close i.e., $< 5 - 10\,\mu V$). The d$I$/d$U$-signal was measured using lock-in technique by modulating the bias across the tunnel junction with a modulation amplitude $U_{mod}$ and a frequency of 4777 Hz or 4675 Hz. For the superconducting tunnel spectroscopy measurements the tip was stabilized at a stabilization bias $U_s$ and stabilization current $I_s$ before switching off the feedback loop. For spectroscopy a time constant of 30 ms was used. For the zero-bias d$I$/d$U$ maps it was 10 ms.

Zero-bias d$I$/d$U$ maps were measured in two distinct ways: In constant-height mode the tip was stabilized at the stabilization parameters and the feedback loop was disabled. The tip was then scanned across the surface on a plane parallel to the Mn film, i.e., parallel to the (110) plane of the Ta. The tilt of the sample was corrected using topographic measurements. In multi-pass mode the constant-current topography of each line was first measured at 4 mV (outside the superconducting gap) and then replayed during the zero-bias scan. This allows for scanning of more corrugated surface topographies as compared to constant-height measurements.

### Tight-binding calculations

For the tight-binding calculations we used the Hamiltonian given by Eq. (1). We assumed two-site interactions (hopping term and Rashba-type SOC) up to third nearest neighbors. The used parameters are $\mu = -3.8$, $\Delta = 1$, $t_1 = 1.1$, $t_2 = 1.0$, $t_3 = 0.9$, $\lambda_1 = 0.55$, $\lambda_2 = 0.5$, $\lambda_3 = 0.45$, $JS_{ML} = 4.0$, $JS_{BL} = 2.5$, $JS_{Ta} = 0$. The hopping parameters and the Rashba

parameters decrease with increasing distance. These values were motivated by parameters used in ref. 18. The two different MSH systems have different values of $JS$. The ratio of $JS_{ML}$ and $JS_{BL}$ is chosen to be roughly the same as the ratio of induced magnetic moments in the top Ta layer for the two systems, which is also roughly the same as the average magnetic moment of the two Mn films, as calculated by DFT (see Supplementary Note 7). As in refs. 12,18,21, all properties are projected onto a monoatomic layer that hosts both superconductivity and magnetic moments. This situation can occur in magnetic films with proximity-induced superconductivity or in superconducting substrates with induced magnetic moments. The monoatomic layer offers an appropriate description for both cases. In addition, the parameters $\mu$, $JS_{ML}$, and $JS_{BL}$ have been chosen such that the ML and BL have a different number of nodal points. As shown in our work, a different number of nodal points in the two adjacent TNPSC is not a necessary condition for the emergence of edge states, but it allows to showcase both types of occurring edge states. Further we have used parameters which lead to a small amount of states within the superconducting gap of the pure Ta substrate in accordance with the experimental observations (for more details see Supplementary Fig. S20 and Supplementary Note 11). To summarize, we have chosen our parameters to achieve a good agreement between experimental observations and theoretical calculations, however, the physical effects are independent of the specific parameters. For calculations in the stripe geometry we used a width of 200 unit cells in each stripe. For this width bulk-like properties exist in each stripe. Furthermore, the edges are sufficiently separated to avoid interactions between edge modes. This is ensured by convergence tests of the stripe width (see Supplementary Fig. S21).

### SP-STM simulations

For the SP-STM simulations the spin-polarized generalization[30,31] of the Tersoff-Hamann model[34] was applied. The tunneling current is given by

$$I(\mathbf{r}_T, V_B) \propto n_T \tilde{n}_S(\mathbf{r}_T, V_B) + \mathbf{m}_T \tilde{\mathbf{m}}_S(\mathbf{r}_T, V_B), \quad (2)$$

where $\mathbf{r}_T$ is the position of the tip, $V_B$ the bias voltage, $n_S$ ($n_T$) is the vacuum LDOS of the sample (tip), and $\mathbf{m}_S$ and $\mathbf{m}_T$ are the respective magnetization DOS. The tilde indicates an integration over energy. As a zero-bias STM measurement has only a small AC bias voltage and no DC voltage, we integrate the LDOS in a small energy range of $-0.02t_2 \leq E \leq 0.02t_2$ around the Fermi energy. From the tight-binding calculations the LDOS of the sample is only known at the lattice sites and not in the vacuum at the tip position. For the wave functions, an exponential decay into the vacuum can be assumed and the contributions of all atom sites $\mathbf{r}_i$ to the LDOS at the tip position are summed[31]

$$\tilde{n}_S(\mathbf{r}_T, V) = \sum_i [\tilde{n}_i^\uparrow(V) + \tilde{n}_i^\downarrow(V)] e^{-2\gamma|\mathbf{r}_T - \mathbf{r}_i|} \quad (3)$$

where $\tilde{n}_i^\sigma(V)$ is the integrated LDOS of spin channel $\sigma$ at lattice site $\mathbf{r}_i$ obtained from the tight-binding model and the decay constant $\gamma = \sqrt{\frac{2m\phi}{\hbar^2}}$ with the work function $\phi$. For the magnetization DOS one obtains:

$$\tilde{\mathbf{m}}_S(\mathbf{r}_T, V) = \sum_i \mathbf{e}_i [\tilde{n}_i^\uparrow(V) - \tilde{n}_i^\downarrow(V)] e^{-2\gamma|\mathbf{r}_T - \mathbf{r}_i|} \quad (4)$$

where $\mathbf{e}_i$ is a unit vector denoting the local spin direction at atom site $i$. For the tip we assume a magnetization direction defined by $\mathbf{e}_T$ and a spin polarization given by $P_T = \frac{n_T^\uparrow - n_T^\downarrow}{n_T^\uparrow + n_T^\downarrow}$. We chose an in-plane lattice

constant of 2.87607 Å (see Supplementary Note 7) and a tip height of 4 Å. A value of $\gamma = 1.0$ Å$^{-1}$ was used as the decay constant.

## Spectral function

The spectral function can be calculated from the solutions for the wavefunctions $|\Psi_{n,\mathbf{k}}\rangle$ as

$$A_i(\mathbf{k}, E) = \sum_n \delta(E_{\mathbf{k}} - \epsilon_{n,\mathbf{k}}) \langle i|\Psi_{n,\mathbf{k}}\rangle \langle \Psi_{n,\mathbf{k}}|i\rangle. \qquad (5)$$

The index $i$ differentiates between the lattice sites, the spin and between electrons and holes. For the calculation of the spectral function in Fig. 7 the contributions of four lattice sites were summed. The spin channels were considered individually and only electronic orbitals were considered.

## Complex band structure

When assuming the wave vector to have a complex value, i.e., $\mathbf{k} = \mathbf{k}_{Re} + \mathbf{k}_{Im} = \mathbf{k}_{Re} + i\kappa$, the Bloch theorem has the form

$$\Psi_{n,\mathbf{k}}(\mathbf{r}) = \Psi_{n,\mathbf{k}_{Re},\kappa}(\mathbf{r}) = \underbrace{e^{i\mathbf{k}_{Re}\cdot\mathbf{r}}}_{\text{periodic}} \cdot \underbrace{e^{-\kappa\cdot\mathbf{r}}}_{\text{decaying}} \cdot u_{n,\mathbf{k}}(\mathbf{r}). \qquad (6)$$

Solving the Schrödinger equation

$$\hat{H}\Psi_{n,\mathbf{k}_{Re},\kappa} = E(\mathbf{k}_{Re},\kappa)\Psi_{n,\mathbf{k}_{Re},\kappa} \qquad (7)$$

therefore leads to $E(\mathbf{k}_{Re}, \kappa)$. Taking $\mathbf{k}_{Re}$ as a parameter and inverting $E$ yields $\kappa(E, \mathbf{k}_{Re})$. The calculation unit cell contained 200 magnetic unit cells in the [1$\bar{1}$0]-direction and one in the [001] direction. In both directions periodic boundaries were assumed. Now $\mathbf{k}_{Re}$ was only varied along [001], leading to $k_{[001]}$. Due to the large unit cell in the [1$\bar{1}$0] direction, the dispersion is nearly flat in this direction and can be neglected. $\kappa$ is assumed to decay perpendicular to the surface, leaving $\kappa_{\|}$[1$\bar{1}$0]. The inversion of the function $E(\mathbf{k}_{Re}, \kappa)$ was performed numerically. For a given value of $\kappa$ the characteristic decay length is given by its inverse $\tau = \frac{1}{\kappa}$. This is the distance after which the signal decays to a value of $\frac{1}{e}$.

## Data availability

The STM data generated in this study have been deposited in the Zenodo database under accession code https://doi.org/10.5281/zenodo.19468148. All details to generate the theoretical data are provided in the manuscript. Other data that support the findings of this study are available from the corresponding authors upon reasonable request.

## Code availability

The code that was employed in this study is available from the authors on reasonable request.

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

## Acknowledgements

K.v.B. gratefully acknowledges financial support from the Deutsche Forschungsgemeinschaft (DFG, German Research Foundation) via pro-jects no. 402843438 and no. 552644472. S.H. gratefully acknowledges financial support from the DFG via project no. 555842692. R.W. acknowledges funding by the European Research Council (ERC Advanced Grant No. 786020). We thank André Kubetzka for his technical support. We acknowledge financial support from the Open Access Publication Fund of Universität Hamburg.

## Author contributions

F.Z. prepared the samples, performed the experiments, and together with K.v.B. and R.L.C. analyzed the data. F.N. developed the tight-binding model, performed the tight-binding calculations and STM simulations and analyzed the data together with S.H. T.D. performed the DFT calculations. F.Z., F.N., R.L.C., S.H., and K.v.B. wrote the manuscript. All authors (F.Z., F.N., R.L.C., T.D., R.W., S.H., and K.v.B.) discussed the data and contributed to the final manuscript.

## Funding

## Competing interests

The authors declare no competing interests.
