## [Peer Review File · Nature Communications]

Spin-polarized edge modes between different magnet-superconductor-hybrids

Corresponding Author: Dr Kirsten von Bergmann

Version 0:

Reviewer comments:

Reviewer #1

(Remarks to the Author)

In this manuscript, the authors performed combined experimental and theoretical study on antiferromagnetic magnet-superconductor hybrid heterostructure (MSH). They deposited monolayer/ bilayer Mn atoms on superconducting Ta(110), and found spin polarized edge states residing at the boundary between the monolayer Mn and bilayer Mn step. By performing tight binding calculation, the authors suggest that the origin of the spin-polarized edge states is from the different topological nodal point superconducting states at the monolayer and bilayer Mn region.

Overall, I feel that the experimental work is of good quality and the theoretical discussion is solid. However, what I find missing is the novelty from the manuscript. Although the paper is technically sound, I was not convinced that this manuscript brings significant advancement in the fundamental understanding of MSH or provides new insights in engineering MSHs with different topological signatures. It is also not clear to me how the observation of these spin-polarized edge states “enables advanced functional design for the exploitation of MSHs as a platform for topology-based applications”. In addition, I have a few concerns regarding the technical aspect of the paper. As a result, I cannot recommend this paper to Nature Communication. My detailed comments are below:

1. What is the condition to have an edge state between two topological nodal superconductors? In other topological systems, such as quantum Hall states, an edge state separates two topological states with distinct topological numbers. What about the topological nodal superconductors? From the discussion in Fig. S17 and Fig. S18, it appears to me that any small change in the material parameters could move the position of the nodal points and hence create an edge state connecting them. In that case, what is the significance of really seeing the edge states at the boundary?
2. The band structure of monolayer Mn on Ta(110) in Figure 6 appears to be quite different from the band structure of a similar structure (monolayer Mn on Nb(110)) in a paper published by some of the same authors calculated by the same model (Ref 18). What are the differences in the parameter choice in the two cases and how sensitive/ robust is the topological nodal point superconductivity to material parameters?
3. I am not entirely convinced that the experimentally observed edge state is located at the ML. Fig S6 shows the z profile and the line dI/dV . The additional broadening in z profile is likely due to the finite width of the tip. In other words, majority of the tunnelling current should still go into the last row of BL Mn in the transition region, where a high dI/dV is observed. In addition, Fig. S7 shows the comparison of topography (left) and constant height image (right) of the same area, where the enhanced edge state is clearly within the BL region. I would suggest the authors to perform better analysis of their data, or focusing on areas where ML/BL Mn steps coincide with the Ta atomic step to better support their argument.
4. The authors mainly worked on edge states along the [001] (FM) direction. They also proposed that similar spin-polarized edge states along the [11-0] (Zigzag) direction. Is such an edge state observed experimentally? A section of [11-0] edge is shown in Fig. 4b, but I do not see any signature of edge state.
5. Are the spin-polarization of the edge states topological in nature, or just a byproduct of the AFM magnetic structure at the sample edge? What is the significance that the state is spin polarized?
6. The authors should provide a more detailed discussion of how these spin-polarized edge states could be used for advanced functional design to demonstrate the potential impact of their work. In particular, since these edge states are always supported by a superconducting substrate, the authors should discuss how they can be detected with other experimental methods and being used in complex circuits.

Reviewer #2

(Remarks to the Author)

In this manuscript, Felix et al. reported the observation of spin-polarized edge modes between two magnet-superconductor hybrids (MSH), which was explained as spin-polarized chiral edge modes that connect topological nodal points of the two different MSHs with different topological superconducting phases, based on tight-binding model calculations. While I am not an expert in theoretical part and thus cannot evaluate the validity of the theoretical section, I have several concerns regarding the experimental claims.

1. What is the evidence for topological nodal point superconductivity in Mn ML and BL grown on Ta (110) surface? The manuscript observes suppressed superconducting gaps with lifted zero-energy DOS in both MSHs, but how does this definitively demonstrate a topological nodal-point phase?
2. How to prove a topologically nontrivial origin of the observed edge modes experimentally? While the authors did extensive theoretical modeling to support this point, how do the authors ensure that the experimental system resides in the same parameter space as the calculations? Given the tunability of theoretical parameters, what independent experimental evidence confirms the nontrivial nature of the edge modes?
3. Could the observed edge modes arise from impurity states (Yu-Shiba-Rusinov states)? The step edges of Mn ML and BL, as shown in Figs. 2a, 3a, S1, S2, and S4a, are not very sharp and contain amounts of defects and disorders. Might the variations in atomic/spin configurations of step edges induce YSR states, which can also explain the spatial inhomogeneity and spin-polarized signal of the edge modes?
4. In Figs. 2b and 3a, it seems likely that the spin-polarized signal is significantly stronger at the island edges than in the interior, and exhibits a strong contrast reverse between the up and down edges. What is the reason of this effect? Is it consistent with the proposed topological scenario?
5. In the theoretical calculations, topological edge modes were predicted along the [110] direction for Mn ML but not BL. Are there consistent experimental observations?

Based on the above questions, I do not think the current version of this manuscript conclusively establish the topological nontrivial origin of the spin-polarized edge states. More rigorous experimental evidence, or exclusions of alternative possibilities (e.g., YSR states), are needed. Therefore, I cannot recommend the publication of this manuscript in Nature Communications.

Reviewer #3

(Remarks to the Author)

In this work, F. Zahner and co-authors report on the combined experimental and theoretical investigation of the properties of the systems consisting of different magnet-superconductor-hybrids. They claim that the signatures of spin-polarized chiral edge modes are observed at the boundary between a Mn-monolayer and a Mn-bilayer on a superconducting Ta surface, supported by their theoretical modeling based on a tight-binding model. This study may be worth publishing in Nat. Commun., however, it is not easy to make this determination from the present version of this manuscript. Below I give several questions and comments which should be properly answered before I can make a decision on the paper.

1, In their tight-binding model, they map all properties onto a single superconducting layer and assume the effective Hamiltonian given in Eq.(1). The parameters such as J_S 'are motivated by the magnitude of the induced magnetic moments in the Ta layer as calculated by density functional theory'. However, this model seems to contradict their measurement because they show the properties of the normal Mn layers by an STM tip, not the properties of the superconducting Ta layers. In my opinion, to map all properties onto the top Mn layer is reasonable. Can the authors improve to clarify the validity of their model?

2, The author claims that they observe the topological edge modes at the boundaries between regions hosting two different topological superconducting phases. It is well-known that trivial Andreev bound states can also arise between two superconductors even at zero phase difference. Can the author distinguish their observed states from the non-topological Andreev bound states between superconductors?

3, Related to the last comment, from the spectra of the ML, the BL, and the oppositely magnetized [001] edges as shown in Figure 4c, I agree that there are subgap edge modes. But the subgap densities of states of the ML and the BL are highly asymmetric. It looks like the trivial Yu-Shiba-Rusinov states stemming from the exposed magnetic atoms at the boundary. Can the author explain the origin of such asymmetry and distinguish their observed states from the non-topological Yu-Shiba-Rusinov states?

4, The authors use ribbon geometry for their computational setup where the stripes are periodic in [001] direction and alternating in [110] direction, as sketched in Figs.5c and 5d. However, the actual experiment has no such periodicity and such assumption could cause the coupling between edge modes at different boundaries in their calculations. Moreover, it seems to me that the authors did not use the periodic ribbon geometry in their previous theoretical investigations [18]. Can the authors explain why they use such modeling particularly in this work?

Minor point:

It seems that there is something wrong with referencing Ref.[21]. There are question marks in the citations.

Version 1:

Reviewer comments:

Reviewer #1

(Remarks to the Author)

The authors' response answered most of my technical questions to satisfaction. They have also improved the manuscript. I have no further questions and can recommend for publication

Reviewer #2

(Remarks to the Author)

I have carefully reviewed the comments from the three reviewers and the authors' responses. Unfortunately, I still have reservations regarding the publication of this manuscript in Nature Communications.

Firstly, the manuscript fails to provide compelling experimental evidence supporting the existence of topological nodal point superconductivity and topological edge states. I acknowledge the authors' statement that "The rigorous experimental identification of a topological origin of edge modes in gapped or nodal point superconductors is up to date still an open issue," and I agree that such experimental identification is indeed challenging. However, it remains essential to experimentally exclude other possibilities—such as YSR states and Andreev bound states, as pointed out by not just one reviewer—as a crucial step in confirming the topological origin. These alternative states possess relatively clear and distinguishable signatures that differ from those of topological edge modes. Although the authors have included additional discussions in the revised manuscript, they have not provided more exclusive experimental evidence to rule out these alternatives.

Secondly, the approach of combining experimental observations with theoretical simulations to confirm topologically nontrivial states has encountered numerous issues in the field of topological superconductivity. While theoretical support is certainly necessary, I believe that the field would benefit more from stronger and more definitive experimental evidence—whether from a confirmatory or an exclusionary perspective.

Reviewer #3

(Remarks to the Author)

The authors have convincingly addressed my concerns and implemented corresponding revisions. They have added more information on their experiment and extended the theoretical discussions. I thus recommend this manuscript for publication in its current form.

Version 2:

Reviewer comments:

Reviewer #2

(Remarks to the Author)

I have carefully read the authors' response to the referees' previous comments regarding the origin of edge states. The authors have made efforts to clarify the physical origin of these edge states, and their argumentation is prudent and rigorous. Moreover, considering the experimental difficulty as well as the good consistency between the experimental data and the theoretical calculations presented in this manuscript, I recommend publication, and it is hoped that there will be more supporting experimental evidence in the future.

REPLY TO REVIEWER COMMENTS

Reviewer 1 (Remarks to the Author):

In this manuscript, the authors performed combined experimental and theoretical study on antiferromagnetic magnet-superconductor hybrid heterostructure (MSH). They deposited monolayer/ bilayer Mn atoms on superconducting Ta(110), and found spin polarized edge states residing at the boundary between the monolayer Mn and bilayer Mn step. By performing tight binding calculation, the authors suggest that the origin of the spin-polarized edge states is from the different topological nodal point superconducting states at the monolayer and bilayer Mn region. Overall, I feel that the experimental work is of good quality and the theoretical discussion is solid. However, what I find missing is the novelty from the manuscript. Although the paper is technically sound, I was not convinced that this manuscript brings significant advancement in the fundamental understanding of MSH or provides new insights in engineering MSHs with different topological signatures. It is also not clear to me how the observation of these spin-polarized edge states “enables advanced functional design for the exploitation of MSHs as a platform for topology-based applications”. In addition, I have a few concerns regarding the technical aspect of the paper. As a result, I cannot recommend this paper to Nature Communication. My detailed comments are below:

A: We thank the Reviewer for assessing our work and for the positive evaluation that ‘the experimental work is of good quality and the theoretical discussion is solid’. Based on the reviewer’s valuable comments and questions we have significantly revised and improved our manuscript. However, we respectfully disagree with the reviewer’s statement that novel aspects are missing in our work. We would like to clarify below what in our view constitutes the novelty of our manuscript.

So far, topological nodal point superconducting (TNPSC) phases have mainly been investigated in the context of boundaries between topological and trivial regions. In contrast to strong topological superconductors, TNPSCs cannot be characterized by a single topological invariant such as the Chern number. Instead, the properties of such systems depend on the topology of the nodal points. Therefore, it is a priori not clear what to expect for a boundary between two distinct TNPSC phases.

In our work, we experimentally observe – for the first time (to the best of our knowledge) – the emergence of edge modes between two TNPSCs. Intriguingly, our tight-binding model calculations show the occurrence of edge modes even for small differences between the two TNPSC regions, even when the number of nodal points is identical in both TNPSCs.

Additionally, we observe a spin polarization of an edge mode, which, to our knowledge, has also not been reported before. The experimental observation of this spin polarization requires a spin-polarized tip, while the investigation of superconducting properties are typically undertaken with superconducting tips. Our theoretical analysis links this spin polarization to the bulk properties of the two adjacent TNPSCs via the complex band structure which has so far also not been investigated for MSH systems. Accordingly, if the electronic structure of one TNPSC domain can be changed continuously — for example, by an applied electric field — the properties of the edge state can also be tuned.

In the revised version of the manuscript we have emphasized these points. Below we provide a point-by-point answer to all specific queries raised by the Reviewer.

Q: 1. What is the condition to have an edge state between two topological nodal superconductors? In other topological systems, such as quantum Hall states, an edge states separate two topological states with distinct topological numbers. What about the topological nodal superconductors? From the discussion in Fig. S17 and Fig. S18, it appears to me that any small change in the material parameters could move the position of the nodal points and hence create an edge state connecting them. In that case, what is the significance of really seeing the edge states at the boundary?

A: For gapped superconducting systems, a Chern number can be calculated that defines the topological phase. This is not possible for nodal point superconductors, as the closed band gap connects occupied and unoccupied states,

preventing the definition of a Chern number. Here, the topology is instead linked to the properties of the nodal points.

In contrast to gapped topological superconductors, edge modes in topological nodal point superconductors may appear only on specific edges, depending on the symmetry of the system and the configuration of the nodal points. Previous studies (e.g., Refs. 12 and 18) have shown that, at the interface with a trivial superconductor, the nodal points are connected by edge modes.

To the best of our knowledge, the interface between two different topological nodal point superconductors has not been systematically investigated before. The absence of a topological invariant describing the systems on either side of the boundary makes a straightforward prediction of the resulting physics challenging. In this work, we investigate for the first time such an interface and find that edge modes also emerge in this case.

We further observe that the formation of edge modes is remarkably robust and occurs even for small differences in the parameters of the two systems (Figs. S17 and S18). This behavior contrasts with that of gapped superconductors, where the system must be tuned into a different phase (with a distinct Chern number) to allow edge modes to form. Since edge modes are experimentally measurable fingerprints of topological superconductivity, future studies could explore how the zero-bias intensity evolves with varying distance between the NPs of adjacent TNPSCs. This may lead to a continuous tuning of the number of states of the edge mode, and therefore also a change of the width of these one-dimensional spin-polarized channels. We believe that our findings will motivate further investigations of interfaces between distinct topological nodal point superconductors, contributing to a deeper understanding of the underlying principles.

In the revised version of the discussion section, we have elaborated more on these aspects:

'Interestingly, edge states emerge even for small differences in the coupling strengths and also for the same number of nodal points in both MSHs. The only requirement for an edge mode between two TNPSC is the presence of NPs in the projected band structure for both MSH systems, so that the different NPs can be connected for the specific direction of the boundary. Consequently, depending on the symmetry and the MSH properties, such boundary modes can arise in several directions or be confined to one direction only. The weight of the edge state in the LDOS, which is a measurable fingerprint, scales with the distance of the connected nodal points. The emergent different electronic structures of the two adjacent MSH systems directly lead to different decay lengths of the edge modes, which in turn governs their spin polarization. Both the width of boundary modes and their spin-polarization impact the lateral transport properties of such MSH systems with TNPSC.'

Q: 2. The band structure of monolayer Mn on Ta(110) in Figure 6 appears to be quite different from the band structure of a similar structure (monolayer Mn on Nb(110)) in a paper published by some of the same authors calculated by the same model (Ref 18). What are the differences in the parameter choice in the two cases and how sensitive/ robust is the topological nodal point superconductivity to material parameters?

A: Previous studies (Refs. 18, 12, 21) have shown that TNPSC can generally be expected for AFM magnetic textures due to the combined symmetry of time-reversal and lattice translation. We have obtained the same result, and Fig. S19 presents our findings of nodal point superconductivity for a large part of the parameter phase space. However, the band structure, including the number of nodal points, depends strongly on the set of parameters.

In Ref. 18 the parameters used in the main text were chosen in a way to achieve a good fit with the experimental data. In our manuscript, we also present parameter sets with good agreement to the experimental data in the main text. While the hopping and Rashba-SOC parameters are very similar to those used in the previous investigation (Ref. 18), the chemical potential and the magnetic coupling are different in our case to achieve a better agreement with the experiments.

In addition to the calculations with the two selected parameter sets in our manuscript we also aim to find general features of TNPSCs. To this end we performed continuous variations of parameters (Figs. S16, S17 and S18) to understand the formation of edge modes. Our results are very general for any nodal point superconductors,

independent of the specific parameters.

In the revised version of the manuscript we have emphasized in the introduction and the tight-binding section, that TNPSC is typically found for antiferromagnetic MSH systems, both in previous investigation and in our study (see also phase diagram in Fig. S20). Furthermore, we have added more information regarding the choice of parameters in the methods section.

Q: 3. I am not entirely convinced that the experimentally observed edge state is located at the ML. Fig S6 shows the z profile and the line dI/dV . The additional broadening in z profile is likely due to the finite width of the tip. In other words, majority of the tunnelling current should still go into the last row of BL Mn in the transition region, where a high dI/dV is observed. In addition, Fig. S7 shows the comparison of topography (left) and constant height image (right) of the same area, where the enhanced edge state is clearly within the BL region. I would suggest the authors to perform better analysis of their data, or focusing on areas where ML/BL Mn steps coincide with the Ta atomic step to better support their argument.

A: We thank the reviewer for bringing up this point regarding the experimental determination of the position of the edge mode. Constant-height zero bias maps, as the ones shown in Fig. S7, are not practical to determine the position of highest intensity of the edge mode, because due to the large tip-sample distance in the ML region the signal is basically zero everywhere. Unfortunately, the investigation of the edge mode at buried step-edges between ML and BL (ML/BL boundary which coincides with the Ta atomic step) is not feasible, because these edges are usually not oriented along high-symmetry directions and therefore do not appear in an ordered manner.

To be able to better locate the position of the DL to ML step edge in our measurements we have now performed STM constant-current simulations across such an edge for different tip-sample distances. After comparing our experimental constant-current height profile to the simulation we have revised our estimate of the last row of BL atoms, see new Supplementary Figure S6. Accordingly, Figures 2 and 4 in the main text as well as Supplementary Figure S4 and S6 have been updated to reflect our new understanding. Because some ambiguity remains in the determination of the exact position of the boundary in the experiments, in the revised manuscript we refrain from making statements regarding the location of the edge mode with respect to the ML/BL boundary.

Q: 4. The authors mainly worked on edge states along the $[001]$ (FM) direction. They also proposed that similar spin-polarized edge states along the $[11-0]$ (Zigzag) direction. Is such an edge state observed experimentally? A section of $[11-0]$ edge is shown in Fig. 4b, but I do not see any signature of edge state.

A: The $[1-10]$ edge between BL and ML is very rare, but one example is visible in Supplementary Figure S9b. The horizontal edge at the top of the Mn BL area shows a slightly increased dI/dU signal (i.e. a weak edge mode). However, this is only seen for atomically straight edges, and as soon as the edge is not perfect the contrast disappears, which is why no edge mode is visible in Fig. 4b. In general the growth of both ML and BL dominantly shows well-developed edges only along $[001]$, which we now explicitly mention also at the end of the BL discussion: *'However, due to the sample morphology only the $[001]$ edges of the elongated BL islands to the Ta substrate could be characterized, while the $[1\bar{1}0]$, and $[1\bar{1}1]$ edges are inaccessible.'*

Q: 5. Are the spin-polarization of the edge states topological in nature, or just a byproduct of the AFM magnetic structure at the sample edge? What is the significance that the state is spin polarized?

A: While the edge modes between two nodal points are topologically protected, their spin-polarization is governed by the underlying AFM structure together with the decay properties away from the boundary. In particular, the spin-polarization depends on the different decay of the edge state into the ML and BL. The importance of the decay for the spin-polarization becomes particularly evident at an up-down-edge, where the oppositely magnetized spins at the boundary would quench the spin-polarization in the case of identical decay on either side of the boundary.

The specific decay properties of the ML and BL stem from their respective bulk properties as shown in our work by the complex band structure. Note that the chirality of the edge mode, i.e. its type of dispersion, inverts when the magnetization at the boundary is opposite, together with an inversion of the spin-polarization. We are not aware that the spin-polarization of such edge modes, their spatial decay, and their origin, have been investigated before.

Q: 6. The authors should provide a more detailed discussion of how these spin-polarized edge states could be used for advanced functional design to demonstrate the potential impact of their work. In particular, since these edge states are always supported by a superconducting substrate, the authors should discuss how they can be detected with other experimental methods and being used in complex circuits.

A: In strong topological superconductors, edge modes occur between two domains of distinct topological phases. Designing such domains typically requires detailed knowledge of the underlying topological invariants. In contrast, our results show that the edge modes in TNPSCs can emerge even for slight modifications of the electronic structure, which shift the nodal point positions. This implies that it may be possible to create and control edge modes within a single TNPSC domain by applying a local perturbation, such as an electric field, without the need to engineer different topological phases. The observed spin-polarization of the edge modes further enhances their potential for future applications. In particular, spin-polarized edge currents could, in principle, be exploited for spin dependent transport.

Our investigations demonstrate that the spin polarization of the edge mode is directly linked to the bulk electronic structure. Therefore, by tuning the bulk electronic structure by external parameters it may be possible to tailor not only the spin character but also other properties of the edge state, such as its localization and number of states. The main focus of our work lies in the observation of spin-polarized edge modes in TNPSCs and their origin. While possible application of our findings are not a key point of our manuscript, we have now strongly revised and extended the discussion section to point out the benefits of TNPSC as mentioned above.

Reviewer 2 (Remarks to the Author):

In this manuscript, Felix et al. reported the observation of spin-polarized edge modes between two magnet-superconductor hybrids (MSH), which was explained as spin-polarized chiral edge modes that connect topological nodal points of the two different MSHs with different topological superconducting phases, based on tight-binding model calculations. While I am not an expert in theoretical part and thus cannot evaluate the validity of the theoretical section, I have several concerns regarding the experimental claims.

A: We thank the Reviewer for assessing our manuscript. We address the raised concerns regarding the experimental claims in a point-by-point fashion and have revised our manuscript accordingly.

Q: 1. What is the evidence for topological nodal point superconductivity in Mn ML and BL grown on Ta (110) surface? The manuscript observes suppressed superconducting gaps with lifted zero-energy DOS in both MSHs, but how does this definitively demonstrate a topological nodal-point phase?

A: As the Reviewer states, in the experimental spectra we observe a filling of the SC gap in the interior of the ML and BL regions. This suggests an interaction between the magnetic moments and the superconducting condensate. The signature of TNPSC is a V-shaped density of states around zero-bias, however, our energy resolution is not sufficient to characterize it experimentally.

In a recent publication (Engström et al., Phys. Rev. B 111, 134505 (2025)) a theoretical investigation on how the topology of nodal points can be identified in STM experiments is presented. While they find some criteria under which scattering at an impurity can reflect specific properties related to the topology in reciprocal space, from a practical point of view it is basically impossible. Indeed, the experimental verification of the topological properties of SC in general has been quite challenging and controversial.

Our conclusion regarding TNPSC relies both on previous Ref. [18, 12, 21] and our current tight-binding calculations for different antiferromagnets in a wide parameter space, that typically find TNP in such MSHs. TNPSC can be expected for AFM magnetic textures due to the symmetry under a combination of lattice-translation and time-reversal. Our tight-binding calculations for a large parameter space (Fig. S20) show that either TNPSC or trivial gapped superconductivity can be expected.

In all these theoretical studies, including the present one, the presence of edge modes, that depend on the specific crystallographic direction of a boundary, has been identified as hallmark for TNPSC, which is what we also observe here experimentally.

In the revised manuscript we have improved and extended the discussion regarding our conclusion of the presence of TNPSC.

Q: 2. How to prove a topologically nontrivial origin of the observed edge modes experimentally? While the authors did extensive theoretical modeling to support this point, how do the authors ensure that the experimental system resides in the same parameter space as the calculations? Given the tunability of theoretical parameters, what independent experimental evidence confirms the nontrivial nature of the edge modes?

A: The rigorous experimental identification of a topological origin of edge modes in gapped or nodal point superconductors is up to date still an open issue. This point is intimately linked to the previous question, as the emergence of TNPSC, and the presence of edge modes connecting the NPs at boundaries, comes hand-in-hand.

In the introduction and in the section 'Experimental results on the Mn ML and Mn BL system' of the revised manuscript we have added new paragraphs discussing the possible origins of enhanced signal at boundaries in more detail and providing a more thorough discussion regarding our conclusion of TNPSC as the origin of the edge modes.

Q: 3. Could the observed edge modes arise from impurity states (Yu-Shiba-Rusinov states)? The step edges of Mn ML and BL, as shown in Figs. 2a, 3a, S1, S2, and S4a, are not very sharp and contain amounts of defects and disorders. Might the variations in atomic/spin configurations of step edges induce YSR states, which can also explain the spatial inhomogeneity and spin-polarized signal of the edge modes?

A: In recent studies (Li et al., Nature Commun. 15, 10121 (2024); Cuperus et al., npj Quantum Mater. 10, 51 (2025)) it was found that the coupling between a magnet and a superconductor can be substantially different in the interior of an island compared to that at its edges. More specifically, this observation was made in a van-der-Waals material, in which the bonding is covalent. This type of directional bonding can lead to significant restructuring and therefore different properties at free edges as compared to the bulk, including the emergence of YSR states at island edges, which were found to be absent in the interior of the island in the mentioned publications.

In the case of our sample system of Mn/Ta(110) we have metallic bonds, which are much less directional and step edges typically do not lead to substantial atom relaxations. In contrast to the mentioned van-der-Waals system we also observe in-gap states not only at edges, but also within the interior of the magnetic islands, demonstrating a coupling of the magnetic moments and the Cooper pairs in the entire MSH system. While we cannot rule out that there is a slight change of atom positions or the size of the magnetic moments between bulk and edges, we expect that this cannot be the origin of the strong zero-bias intensity we see in our experiments.

We would like to note that many of the [001] edges of the ML to the Ta are rather straight, see e.g. the step edge indicated by the red arrow in Fig. 2a. However, other edge directions have more kinks and sometimes appear round. In contrast, nearly all [001] edges of the BL to the ML are very straight, interrupted only by single atomic kinks, where an additional row of atoms is attached to the island. In our opinion also the atomically well-defined structure of the edges rules out that spatial inhomogeneity could be the origin of the increased intensity at the edges.

As stated in our answer to the previous question, in the revised manuscript we have discussed the possible origins of enhanced signal at boundaries, including YSR-states, in more detail and now provide a more thorough discussion regarding our conclusion of TNPSC as the origin of the edge modes.

Q: 4. In Figs. 2b and 3a, it seems likely that the spin-polarized signal is significantly stronger at the island edges than in the interior, and exhibits a strong contrast reverse between the up and down edges. What is the reason of this effect? Is it consistent with the proposed topological scenario?

A: The current map in Fig. 2b is measured at a bias voltage of +5 mV, i.e. outside the superconducting gap. It reflects the magnetic signal obtained in a SP-STM measurement: the fine lines reveal the AFM order; the strong dark and white appearance of the upper and lower step edge to the Ta is an artifact of the measurement due to the feedback loop which is typical for current maps (fast scan direction from bottom to top).

Also Fig. 3a is measured outside the superconducting gap at +5mV, i.e. no properties related to the superconductivity appear. Here, the fast scan direction is from the right side to the left. Due to the feedback loop artifact only step edges that are equivalent with respect to the fast scan axis can be compared quantitatively. In our case, we focused on the step edges on the right side of the BL islands, as indicated by the black rectangles. The reviewer is correct, that these step edges in general have a higher intensity than the interior of the islands. This is due to the mentioned feedback artifact in the current map. For this reason, we do not quantitatively compare the signals between the island bulk and the edge, but only those between different edges. Note that also due to a possible asymmetric tip only the same-side edges can be compared quantitatively.

We have added more information on this kind of artifact in Fig.2 and Fig.3 and in the methods section to prevent misunderstandings in the data interpretation.

Q: 5. In the theoretical calculations, topological edge modes were predicted along the [110] direction for Mn ML but not BL. Are there consistent experimental observations?

A: For bcc(110) surfaces, typically the growth leads to long straight edges along [001], which we also find in our system. Other edge directions are energetically less favored. In ML islands also this edge direction to the Ta is dominant, but for submonolayer samples also other directions occur as shown in Fig. 2, however, no straight [1-10] edges have been observed.

The BL islands grow on the ML and therefore boundaries between the BL and the Ta substrate are rare. One example is shown in Fig. S8, where a [001] edge of the BL is next to Ta; such an edge along [1-10] has not been observed and consequently an experimental characterization is not possible.

We have added this information in the revised version of the manuscript in the discussion related the experiments on the BL: *'However, due to the sample morphology only the [001] edges of the elongated BL islands to the Ta substrate could be characterized, while the $[1\bar{1}0]$, and $[1\bar{1}1]$ edges are inaccessible'*

Q: Based on the above questions, I do not think the current version of this manuscript conclusively establish the topological nontrivial origin of the spin-polarized edge states. More rigorous experimental evidence, or exclusions of alternative possibilities (e.g., YSR states), are needed. Therefore, I cannot recommend the publication of this manuscript in Nature Communications.

A: Based on the reviewer's questions we have revised the manuscript and significantly extended the discussion on possible origins of boundary states and our arguments for TNPSC. As discussed above, we cannot provide rigorous experimental evidence to establish the topologically non-trivial origin of the spin-polarized edge states, but rather rely on comparison to tight-binding calculations. However, we would like to emphasize that our theoretical study has for the first time addressed the properties of the boundary between two TNPSCs and revealed the existence of edge modes and also the origin of their spin-polarization and their spatial decay into the two TNPSCs. The experimental data of the Mn BL and Mn ML exhibit all the features that are expected for such TNPSCs, including the spin-polarization of edge modes between them. Therefore, we believe that our revised manuscript is suitable for publication in Nature Communications.

Reviewer 3 (Remarks to the Author):

In this work, F. Zahner and co-authors report on the combined experimental and theoretical investigation of the properties of the systems consisting of different magnet-superconductor-hybrids. They claim that the signatures of spin-polarized chiral edge modes are observed at the boundary between a Mn-monolayer and a Mn-bilayer on a superconducting Ta surface, supported by their theoretical modeling based on a tight-binding model. This study may be worth publishing in Nat. Commun., however, it is not easy to make this determination from the present version of this manuscript. Below I give several questions and comments which should be properly answered before I can make a decision on the paper.

A: We thank the Reviewer for evaluating our manuscript and provide below a point-by-point answer to all questions and comments based on which we have improved the clarity of our manuscript.

Q: 1, In their tight-binding model, they map all properties onto a single superconducting layer and assume the effective Hamiltonian given in Eq.(1). The parameters such as JS ‘are motivated by the magnitude of the induced magnetic moments in the Ta layer as calculated by density functional theory’. However, this model seems to contradict their measurement because they show the properties of the normal Mn layers by an STM tip, not the properties of the superconducting Ta layers. In my opinion, to map all properties onto the top Mn layer is reasonable. Can the authors improve to clarify the validity of their model?

A: In previous investigations of two-dimensional MSHs only one system was investigated at a time. In those cases, it is somewhat irrelevant whether one uses a picture where novel properties emerge from magnetic moments that are induced in the superconducting material, or a picture where a magnetic film becomes superconducting due to the proximity effect. In those previous studies just a single JS is used, since only one magnetic-superconductor interface is present.

Our study is the first to investigate the interface between two distinct TNPSCs, which requires two sets of parameters to capture the different regions. Since determining exact system parameters is not straightforward, we based our choice partly on previous studies of related MSH systems. In contrast to previous studies we have two distinct regions, each represented by a unique value of JS .

In response to the Reviewer’s question, and because our main findings of the emergence of spin-polarized edge modes at the boundary between two different TNPSC do not rely on the specific set of parameters, in the revised manuscript we put less emphasis on the motivation of the JS values by DFT calculations. However, we assume that the main contribution to the properties should originate from the interface between the superconductor and the magnetic film. In the revised methods section we mention both perspectives on MSH systems: the superconductor with magnetic moments, and the magnet that is superconducting:

‘The two different MSH systems have different values of JS . The ratio of JS_{ML} and JS_{BL} is chosen to be roughly the same as the ratio of induced magnetic moments in the top Ta layer for the two systems, which is also roughly the same as the average magnetic moment of the two Mn films, as calculated by DFT (see Supplementary Note 7). As in Refs. [18, 12, 21], all properties are projected onto a monoatomic layer that hosts both superconductivity and magnetic moments. This situation can occur in magnetic films with proximity-induced superconductivity or in superconducting substrates with induced magnetic moments. The monoatomic layer offers an appropriate description for both cases. In addition, the parameters μ , JS_{ML} , and JS_{BL} have been chosen such that the ML and BL have a different number of nodal points.’

Q: 2, The author claims that they observe the topological edge modes at the boundaries between regions hosting two different topological superconducting phases. It is well-known that trivial Andreev bound states can also arise between two superconductors even at zero phase difference. Can the author distinguish their observed states from the non-topological Andreev bound states between superconductors?

A: The rigorous experimental identification of a topological origin of edge modes in gapped or nodal point super-

conductors is to date, still an open issue. Indeed, the experimental verification of the topological properties of SC in general has been quite challenging and controversial.

In the introduction of the revised manuscript we have added one paragraph to discuss the possible origins of enhanced signal at boundaries in more detail, specifically trivial Andreev bound states and trivial YSR states. We now also provide a more thorough discussion regarding our conclusion of TNPSC as the origin of the experimentally observed edge modes, in a new paragraph added to the section 'Experimental results on the Mn ML and Mn BL system'.

However, Andreev reflections typically arise in lateral transport experiments with a current flow perpendicular to the interface. While in an STM measurement a peak arising in the DOS at a structural step edge in a high- T_C superconductor was interpreted as an Andreev bound state (Misra et al., PRB 66, 100510(R) (2002)), we have no reason to believe that they play a role in our system. Based on the good agreement of experimental results and the calculations, we consider TNPSC as the most likely scenario. Our interpretation of the experimental edge modes originating from a TNPSC phase relies on three points. Firstly, AFM structures in general can induce TNPSC, see Ref.[12]. Secondly, a TNPSC phase is known to arise in a quite similar system (Ref. [18]). Lastly we find a very good agreement with our theoretical simulations, which assume a TNPSC phase.

Q: 3, Related to the last comment, from the spectra of the ML, the BL, and the oppositely magnetized [001] edges as shown in Figure 4c, I agree that there are subgap edge modes. But the subgap densities of states of the ML and the BL are highly asymmetric. It looks like the trivial Yu-Shiba-Rusinov states stemming from the exposed magnetic atoms at the boundary. Can the author explain the origin of such asymmetry and distinguish their observed states from the non-topological Yu-Shiba-Rusinov states?

A: As mentioned in context with the last question we have added a more detailed discussion on the possible origin of boundary states, including YSR-states. We would like to note that the experimentally measured dI/dU signal in the gap is related to the spectral function. While the TNPSC band structures are symmetric around the Fermi-energy, the spectral function, which is related to the signal measured in tunnel spectroscopy, is not, see for instance Fig. 7b, where several bands below the Fermi-energy have a significantly stronger weight as compared to their counterparts above the Fermi-energy. Consequently, the measured differential tunnel conductance can be asymmetric. Indeed, also the spectra in the interior of the island show a similar asymmetry, see Fig. 1b.

Q: 4, The authors use ribbon geometry for their computational setup where the stripes are periodic in [001] direction and alternating in [110] direction, as sketched in Figs.5c and 5d. However, the actual experiment has no such periodicity and such assumption could cause the coupling between edge modes at different boundaries in their calculations. Moreover, it seems to me that the authors did not use the periodic ribbon geometry in their previous theoretical investigations [18]. Can the authors explain why they use such modeling particularly in this work?

A: In previous investigations, often quasi-1D ribbons were used. These are periodic, e.g., in the [001] direction and finite in the perpendicular direction (here [1-10]). The infinite length in one direction allows to calculate a dispersion relation, which gives a good understanding about the electronic structure. Outside of the ribbon there exist no atoms, creating a vacuum. This vacuum is a trivial state and consequently edge modes between nodal points can form at the boundary of the TNPSC and the vacuum. In our investigation, we are interested in edge modes occurring between two different nodal point superconductors.

Consequently, we have to use two regions with atoms, albeit with different properties. The periodic stripe geometry ensures that only interfaces between the two TNPSC of interest exist, and that in a projection onto the boundary no additional edge modes to trivial states appear.

The reviewer is right, that in this setup a hybridization between edge modes across one stripe can occur. As the edge modes decay exponentially (c.f. Fig.8 and Fig. S14) the hybridization will decrease exponentially with the width of the ribbon. For our calculations we have performed convergence tests regarding the width of the stripes,

to make sure that such kind of hybridization is absent.

In the revised manuscript we have added a comment in the methods section and also included a new Supplementary Figure S21 to show the convergence tests.

Q: Minor point: It seems that there is something wrong with referencing Ref.[21]. There are question marks in the citings.

A: We thank the reviewer for pointing this out and have updated this reference.

We hope that our answers to the reviewer's queries and the improvements to the manuscript have convinced the reviewer of the validity of our conclusions and the significance of our work.

REPLY TO REVIEWER COMMENTS

Reviewer #1 (Remarks to the Author):

The authors' response answered most of my technical questions to satisfaction. They have also improved the manuscript. I have no further questions and can recommend for publication.

We thank the Reviewer for evaluating our manuscript and recommending it for publication.

Reviewer #2 (Remarks to the Author):

I have carefully reviewed the comments from the three reviewers and the authors' responses. Unfortunately, I still have reservations regarding the publication of this manuscript in Nature Communications.

Firstly, the manuscript fails to provide compelling experimental evidence supporting the existence of topological nodal point superconductivity and topological edge states. I acknowledge the authors' statement that "The rigorous experimental identification of a topological origin of edge modes in gapped or nodal point superconductors is up to date still an open issue," and I agree that such experimental identification is indeed challenging. However, it remains essential to experimentally exclude other possibilities—such as YSR states and Andreev bound states, as pointed out by not just one reviewer—as a crucial step in confirming the topological origin. These alternative states possess relatively clear and distinguishable signatures that differ from those of topological edge modes. Although the authors have included additional discussions in the revised manuscript, they have not provided more exclusive experimental evidence to rule out these alternatives.

Secondly, the approach of combining experimental observations with theoretical simulations to confirm topologically nontrivial states has encountered numerous issues in the field of topological superconductivity. While theoretical support is certainly necessary, I believe that the field would benefit more from stronger and more definitive experimental evidence—whether from a confirmatory or an exclusionary perspective.

We thank the Reviewer for again assessing our manuscript. Based on the Reviewer's comment we have again revisited the properties of possible in-gap features.

The Reviewer agrees that an experimental identification of a topological origin of edge modes is challenging. In view of this, the Reviewer suggests to experimentally exclude other possibilities, and states that the alternative states (YSR and Andreev bound states) possess relatively clear and distinguishable signatures that differ from those of topological edge modes. However, as we detail below, we are not convinced that for our system there are such fingerprints that would allow us to rigorously rule out alternative explanations.

YSR states of single spins on surfaces can occur anywhere in the gap depending on the coupling strength of the spin to the superconductor (SC), the size of the spin, and magnetic anisotropies. They appear at particle-hole symmetric energies. Their spectral function, and thus the measured spectrum, may be asymmetric with respect to zero-bias due to different particle and hole weights. In a lattice of single spins the YSR states are independent for larger distances, and they start to hybridize when the distance between the spins becomes smaller, which can result in YSR bands. For a densely packed atomic lattice with a static magnetic ground state, topological SC may arise, which can either be gapped or exhibit nodal points, as for instance in our antiferromagnetic layers. In this

regime of strongly coupled magnetic moments, which form a nodal-point SC phase, in our view the model of YSR states is not applicable. However, one might argue otherwise and claim that YSR states in the interior of the islands may be different from those at edges, e.g. due to a slightly different size of the magnetic moment, due to a slightly varied magneto-crystalline anisotropy, or due to a changed coupling to the SC, giving rise to a difference in the local density of states around zero-bias for the edges compared to the interior of the magnetic islands.

Andreev bound states (ABS) arise at interfaces with at least one SC involved. To our understanding, any electronic state in the vicinity to the superconducting gap may couple to the SC and give rise to in-gap states, which are often summarized under the term ABS. In the context of nanowires in contact to a superconductor, theoretical investigations have elaborated on the differences between ABS and Majorana bound states (MBS), which both appear at zero bias, see Refs. Liu et al., PRB 96, 075161 (2017) and PRB 97, 214502 (2018). From an STM perspective, ABS can arise due to planar boundaries parallel to the surface, as in proximity coupled normal metal films on SC substrates, or at lateral interfaces, i.e. when different materials are separated by a step or some other boundary. As shown by Schneider et al., Nature 621, 60 (2023) and Wang et al., PRL 135, 076201 (2025) the energy of these ABS, which are particle-hole-symmetric, depends on the coupling of the electronic state to the SC. For the cases shown in these two publications, the ABS are induced by Andreev reflections at the planar boundaries between a SC and normal metallic states, and the energies of the in-gap states do not depend on the position within the SC but only on the lateral confinement. In these systems, the ABS are not localized on the edges of the system, but rather present in the interior of the investigated 2D systems.

To obtain in-gap ABS localized on the edges of the islands, an electronic state localized on the island edges would be needed already in the normal metallic state. In principle, any boundary in a solid-state system may induce such localized electronic states. In the case of metallic systems, as ours, such boundary states typically have a width on the order of 100 meV (see e.g. Pietzsch et al., PRL 96, 237203 (2006)), about two orders of magnitude larger than the gap of the SC. This would imply a strong coupling of the ABSs induced by this kind of localized electronic states. For such a strong coupling, the ABS are expected to be energetically located very close to the coherence peaks of the superconducting gap, and certainly far from zero energy. However, in our systems we observe the boundary modes in the superconducting state very close to zero energy. We thus exclude the possibility of ABS as origin for the observed in-gap edge states.

The properties of the topological edge modes in TNPSC depend on the specific boundary (as also YSR states and ABS would). As shown by Bazarnik et al., Nature Commun. 14, 614 (2023) the topological edge modes are flat and do not disperse for edges with alternating magnetic moments (the antiferromagnetic edges along [1-11] of an AFM monolayer on a bcc(110) surface), and consequently also do not show a spin-polarization. However, the edges along the other two high-symmetry directions of a bcc(110) magnet-superconductor hybrid ([001] and [1-10]) exhibit chiral edge modes with a dispersion and particle-hole symmetry, see Bazarnik et al. and also Fig. 6c-e and Fig. 7a of the present manuscript. For a strong dispersion, this leads to a double-peak structure in the density of states as demonstrated by Bazarnik et al., Nature Commun. 14, 614 (2023). Our new TB calculations demonstrate that these chiral edge modes are spin-polarized, see Fig. 7b,c. Depending on the asymmetry of particle- and hole-weight, on the spin-polarization of the edge mode, and that of the used magnetic STM tip, this gives rise to asymmetric spectra, as seen in our data, e.g. Fig. 4c.

To conclude, we rule out ABSs as the origin for the observed edge states. While we cannot completely rule out the contribution of YSR states, many arguments support the interpretation that our boundary modes are chiral edge modes in a topological nodal-point superconductor, last but not least based on the good agreement with the TB calculations. Even if there are some reservations

regarding their proof by the experimental data alone, our calculations clearly reveal the origin and the spin-polarization of boundary modes between different TNPCs.

We have slightly revised the discussion regarding ABSs in the manuscript, adding a reference regarding the merging with coherence peaks in the strong coupling regime. We hope the reviewer agrees with our elaborations and the changes to the manuscript, values our reluctance to over-interpret and -sell our experimental data, and also recommends publication of the revised version in Nature Communications.

Reviewer #3 (Remarks to the Author):

The authors have convincingly addressed my concerns and implemented corresponding revisions. They have added more information on their experiment and extended the theoretical discussions. I thus recommend this manuscript for publication in its current form.

We thank the Reviewer for evaluating our manuscript and recommending it for publication.